# Control of cell morphology and differentiation by substrates with independently tunable elasticity and viscous dissipation

Elisabeth E. Charrier[1,2], Katarzyna Pogoda[1,3], Rebecca G.  Wells[2] & Paul A.  Janmey[1]

The mechanical properties of extracellular matrices can control the function of cells. Studies of cellular responses to biomimetic soft materials have been largely restricted to hydrogels and elastomers that have stiffness values independent of time and extent of deformation, so the substrate stiffness can be unambiguously related to its effect on cells. Real tissues, however, often have loss moduli that are 10 to 20% of their elastic moduli and behave as viscoelastic solids. The response of cells to a time-dependent viscous loss is largely uncharacterized because appropriate viscoelastic materials are lacking for quantitative studies. Here we report the synthesis of soft viscoelastic solids in which the elastic and viscous moduli can be independently tuned to produce gels with viscoelastic properties that closely resemble those of soft tissues. Systematic alteration of the hydrogel viscosity demonstrates the time dependence of cellular mechanosensing and the influence of viscous dissipation on cell phenotype.

[1] Institute for Medicine and Engineering, University of Pennsylvania, 3340 Smith Walk, Philadelphia, PA 19104, USA. [2] Division of Gastroenterology, Department of Medicine, Perelman School of Medicine, University of Pennsylvania, 421 Curie Boulevard, Philadelphia, PA 19104, USA. [3] Institute of Nuclear Physics Polish Academy of Sciences, PL-31342 Krakow, Poland. Correspondence and requests for materials should be addressed to E.E.C. (email: charrier@pennmedicine.upenn.edu)

In vivo, most cells are organized in tissues where they are interconnected with other cells and with the biopolymers forming the extracellular matrix. The homeostasis of tissues is ensured by the ability of cells to sense and respond to their biological and mechanical environments. Most studies of cellular mechanosensing have used purely elastic crosslinked poly-acrylamide gels[1,2] with almost no dissipation of deformation energy (loss modulus). However, real tissues such as brain, liver, spinal cord and fat often have loss moduli that are 10 to 20% of their elastic storage moduli[3–8] over a large range of time scales. A few very soft tissues like brain behave like viscoelastic fluids with no permanent elastic storage modulus, but most biological tissues behave as viscoelastic solids on a time scale relevant to mechanical sensing, in which stress after deformation decays partially but not totally over a period of seconds to minutes[8–16]. In some diseased tissues such as breast tumors, the rate of stress relaxation is altered more than the magnitude of the elastic modulus[10].

Viscoplastic or viscoelastic fluid substrates have been created to study the effect of substrate stress relaxation on cells[11,17–19]. The use of these materials has revealed new cellular behaviors, but the irreversible rearrangement of the materials themselves in response to the forces produced by cells makes it hard to separate the effect of substrate viscosity from the structural reorganization of the matrix, which can lead to local concentration of adhesive ligands. The response of cells to a time-dependent viscous loss in a dissipative solid is largely uncharacterized because appropriate viscoelastic materials are lacking for quantitative studies. Here we report the synthesis of soft viscoelastic solids for which the elastic and viscous moduli can be independently tuned to produce gels with viscoelastic properties that mimic those of soft tissues. This was done by creating permanently crosslinked networks of polyacrylamide (PAA) that sterically entrap but do not bind very high molecular weight linear polymers of PAA. The chemistry of these systems allows cell adhesion ligands such as collagen and fibronectin to be attached exclusively to the crosslinked elastic network, to the viscous linear chains or to both viscous and elastic elements.

## Results

**Entrapping linear PAA in a network forms viscoelastic gels**. PAA is a biologically inert polymer forming hydrogels of variable elasticity that is commonly used as a soft substrate for cell culture[20] after adhesive molecules such as integrin ligands are covalently attached to its surface. Once polymerized, acrylamide and bis-acrylamide form purely elastic gels with time-independent responses to stress. In order to obtain viscoelastic PAA gels, a dissipative element, linear PAA, was included within the structure of the crosslinked gels (Fig. 1a). The mixture of entrapped and slowly relaxing linear chains within the permanently crosslinked elastic network resulted in a viscoelastic gel characterized by a shear storage elastic modulus G' and a significant loss modulus G" (Fig. 1d, e). As expected, G' increased over time during the polymerization of the network. G" also increased during network formation, indicating that the confinement of the linear PAA molecules is the origin of gel viscoelasticity (Fig. 1b). The stress relaxation of these gels showed the stress evolution typical of a viscoelastic solid relaxing to a plateau value after approximately 10 to 100 s (Fig. 1c). The creep function of the gel confirmed a significant viscous creep, while the recovery after stress was removed demonstrated the absence of plasticity as the gel returned to its shape before deformation (Fig. 1g). Our PAA gels differ in this respect from the system reported by Cameron et al.[19]; their partially crosslinked PAA gels keep flowing under the application of a constant stress, which is typical of viscoelastic fluids. The frequency dependence of our viscoelastic gels during low strain oscillatory deformation, tested from $1.59.10^{-3}$ Hz to 10 Hz (Fig. 1f and Supplementary Figure 1b), showed a very weak frequency dependence of G', while G" varied over an order or magnitude in the range of frequencies tested. Purely elastic PAA gels have a constant G' over the range of frequencies tested (Supplementary Figure 1a). In a biological context, the sensitivity of G" to frequency is important: the rate at which cellular mechanisms probe the substrate could influence the amount of dissipation felt by cells. As the value of G' value is frequency independent, the elasticity felt by cells will not depend on the time scale of the cellular machinery involved in mechanosensing. Two studies reported a time scale of 10 s to describe traction force fluctuations exerted by cells on their substrates[21,22]. This finding suggests that a relevant frequency for cell mechanosensing is about 1 rad/s. For this reason, for our gels we report the values of G' and G" measured at the biologically relevant frequency of 1 rad/s (0.16 Hz).

In the first step of the procedure to make viscoelastic gels, very high molecular weight linear PAA was generated by using a minimal concentration of the free radical initiator and maximizing chain propagation by preventing quenching of the growing chain. The viscosity of linear PAA solutions at a range of concentrations was measured to determine the specific viscosity [η] (Supplementary Figure 2a). Based on these results and previous studies that related [η] to molecular weight[23], we estimated the average molecular weight of the linear PAA molecules as 1.6 MDa (Supplementary Methods). We also performed dynamic light scattering measurements and observed that the linear polymers had an average hydrodynamic radius ($R_h$) of about 40 nm, a size consistent with the measured weight of 1.6 MDa and the degree of swelling of PAA in water, which is a good solvent for this polymer[23], which predicted a hydrodynamic diameter of approximately 55 nm (Supplementary Methods). The linear molecules were heterogeneous in size and their hydrodynamic radii were distributed between 20 nm and 400 nm (Supplementary Figure 2b).

In the second step, the viscous linear polyacrylamide solution was mixed with a solution of monomeric acrylamide and bis-acrylamide that polymerized into a PAA network with a mesh size of 10 nm, trapping the entangled linear molecules of PAA within a crosslinked network. Atomic force microscopy (AFM) stiffness mapping was performed on the resulting hydrogels to quantify the impact of the linear PAA on the nano-scale stiffness of the gel. The degree of stiffness heterogeneity is slightly increased for gels entrapping linear PAA in comparison to gels made of a PAA network only (Supplementary Figure 2c). However, the overall stiffness of the two types of gels is similar and the distributions of the local Young's moduli overlapped for about 85% of the local measurements performed (Supplementary Figure 2d). To confirm that the gels maintained their mechanical properties over time, the diffusion of the linear PAA in the network was estimated. The diffusion time D, of linear polyacrylamide with a 40 nm hydrodynamic radius $R_h$ in infinite dilution in a solvent with a viscosity η, calculated from the expression $D = kT/6\pi\eta R_h$, where k is the Boltzmann's constant and T, the temperature, is 5 μm²/s in $H_2O$ and its diffusion in the highly entangled network of crosslinked polymer plus linear polyacrylamide is several orders of magnitude slower. Since even in infinitely dilute solutions the linear polymer would need 2000 s to diffuse 100 μm[24], the diffusion or reptation time of the entangled polymer to escape the network is on the order of days or weeks. This slow diffusion is the reason that even small proteins require a large electric field rather than diffusion to transfer from a polyacrylamide gel during immunoblotting

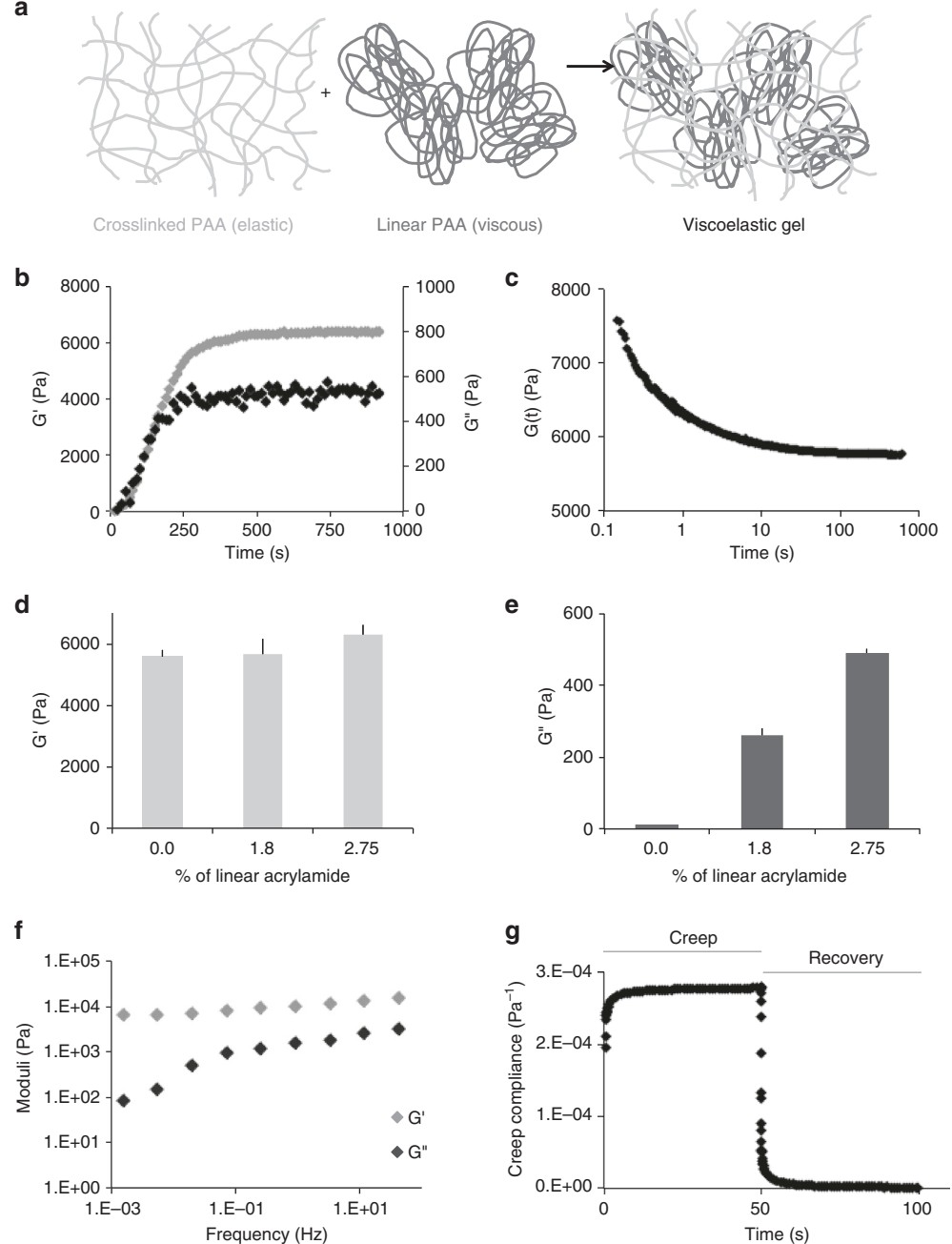

**Fig. 1** Structure and mechanical characterization of the viscoelastic PAA gels. **a** Cartoon representing the association of an elastic network of PAA with a viscous solution of linear PAA to form a viscoelastic gel. Trapping of linear chains of PAA in a network of PAA leads to the formation of a viscoelastic gel. **b** Evolution of G' and G" during the polymerization of a viscoelastic gel: G' and G" both increase during the formation of the branched PAA network. G', light gray (left axis); G", dark gray (right axis). **c** Representative plot of the stress relaxation of a viscoelastic gel containing 2.75% linear polyacrylamide: evolution of the shear modulus over time under a 10% strain. **d** Average value of G' as characterized by a 2% oscillating shear stress at a frequency of 0.159 Hz and 2% strain; $n = 5$ gels per condition. Error bars represent the standard error. **e** Average value of G" as characterized by a 2% oscillating shear stress at the frequency of 0.159 Hz and 2%strain; $n = 5$ gels per condition. Error bars represent the standard error. **f** Values of G' and G" as a function of the oscillation frequency for the highly viscous gel (designed as the G' = 5 kPa, G" = 500 Pa gel). The frequency sweep was performed from 0.005 Hz to 10 Hz. G' shows a weak frequency dependence while G" varies over a decade in the range of frequencies probed. **g** Creep and recovery of the viscoelastic gel containing 2.75% of linear PAA. The creep compliance was followed during the application of a stress of 100 Pa for 50 s, and then the recovery was monitored for 50 s

procedures, and the linear PAA chains here are more than an order of magnitude larger. This expectation was experimentally verified by measuring G' and G" on gels immersed in phosphate-buffered saline (PBS) for a week (Supplementary Figure 3a). The average value of the loss moduli decreased no more than 10% after the gel was immersed in PBS for 7 days.

The amounts of acrylamide, bis-acrylamide and linear PAA were adjusted to generate three kinds of gels with similar elasticity but variable viscosity (see Table 1 and Fig. 1d). These viscoelastic gels were designed to have mechanical properties close to biological tissues, with G" equal to 5 or 10% of G' when measured at low strain and a frequency of 1 rad/s.

| Table 1 Average values of the elastic moduli (G′) and the viscous moduli (G″) of polyacrylamide gels at a frequency of 0.16 Hz, as a function of the percentage of acrylamide, bis-acrylamide and linear PAA ||||||
|---|---|---|---|---|
| **Average G′ (Pa)** | **Average G″ (Pa)** | **% Acrylamide** | **% Bis-acrylamide** | **% Linear acrylamide** |
| 5580 | 10 | 8 | 0.1 | 0 |
| 5660 | 260 | 8 | 0.125 | 1.8 |
| 6280 | 490 | 8 | 0.125 | 2.75 |
| $n \geq 5$ gels |||||

The surface of the gels enables cell adhesion by presenting proteins such as fibronectin and collagen that engage cell receptors; PAA itself is not able to adsorb proteins from the media or to support cell adhesion. Therefore, the amide groups of the PAA must be chemically activated in order to be crosslinked to biologically active molecules. We used this property of the PAA to our advantage, attaching collagen I or fibronectin to the network of PAA, to the linear PAA or to both. Mixing acrylic-acid N-hydroxy-succinimide ester (AA-NHS) with acrylamide monomers introduces active monomers, selectively either within the linear chains or the crosslinked network that can be conjugated with proteins. Incubating AA-NHS with acrylamide during the preparation of the linear PAA resulted in activated linear PAA that can be entrapped within inactive crosslinked PAA networks. Similarly, incubating AA-NHS only with the mixture of acrylamide and bis-acrylamide enables activation of an elastic network that can be polymerized around previously synthesized inactive linear PAA. The chemical activation of the amide groups can also be performed on previously polymerized PAA with sulfosuccinimidyl 6-(4′-azido-2′-nitrophenylamino) hexanoate (sulfoSANPAH). In order to activate all of the PAA (linear chains and network) gels were treated with sulfoSANPAH after the polymerization of the network around the linear chains. The chemical crosslinking of proteins at the surface of gels did not affect their mechanical properties (Supplementary Figure 3b). Thus, the biological signal at the surface of the gel can be selectively coupled to its elastic, viscous or viscoelastic properties.

**Linear PAA does not affect presentation of adhesion protein.** To characterize the density of adhesion molecules through the networked or the linear PAA, AFM was used to probe the adhesion proteins at the surface of the gels. The AFM tip coated with anti-collagen antibodies was approached to the surface of the gel previously functionalized with collagen I. Once the contact between the tip and the collagen was established, the tip was retracted until the contact broke (Fig. 2a). Using this method, the force required for the contact to break is called $F_{adh}$, and the retraction distance at which the contact breaks is $L_{adh}$. Elastic and viscoelastic gels presenting collagen I on the network had similar retraction force and length (Fig. 2b, c). Thus, the accessible collagen density on both types of gel was similar, verifying that presence of linear PAA did not affect the presentation of adhesion proteins on the network of the gels. When the proteins were attached to the linear PAA, the adhesion force decreased and the adhesion length increased, in comparison to when the proteins adhered to the network. The collagen I probed in this condition was linked to a more compliant element than the collagen I presented on the network and could be displaced by forces in the range of the pN even in the short time of the AFM movement. Such forces are in the same range as the forces exerted at a single

integrin adhesion site[25]. These results show that the proteins attached to the network and the linear PAA have different underlying mechanical properties that might affect cell response.

**Viscosity and ligand presentation regulate cell spreading.** NIH 3T3 fibroblasts were plated on elastic and viscoelastic PAA gels with the same shear storage modulus G′ of 5 kPa and shear loss moduli G″ of 0 Pa, 200 Pa or 500 Pa (at 0.16 Hz). The gels were conjugated to collagen I (Fig. 3a), or fibronectin (Fig. 4a), linked to one or both of the two forms of PAA. Bright field images of the fibroblasts were taken 24 h after plating (Figs. 3b and 4b) and used to quantify the cells' average projected areas (Figs. 2c and 3c). On purely elastic 5 kPa gels coated with collagen I or fibronectin, fibroblasts spread similarly as on other stiff surfaces, but their response to viscoelastic gels depended on the magnitude of the shear loss modulus (G″).

3T3 fibroblasts had overall smaller areas on viscoelastic gels than on purely elastic gels when adhesion proteins were presented only on the crosslinked network of PAA. This result shows that cells are sensitive to the viscosity of the underlying substrate even if the adhesion molecules are bound only to the permanently crosslinked elastic component of the gel. On collagen I-coated gels, fibroblast spreading was significantly decreased on both types of viscoelastic gels (G″ = 200 Pa and G″ = 500 Pa) compared to elastic gels, whereas when they were bound to gels through fibronectin, fibroblasts responded to viscosity only for the gel with the largest dissipation (G″ = 500 Pa). The different effect of viscosity when adhesion to the substrate is mediated through collagen I compared to fibronectin suggests that distinct biochemical signals from the different classes of integrins that engage these two ligands determine the threshold sensitivity to viscosity. As shown in Fig. 1f, the viscous modulus of the gels had a strong frequency dependence. Therefore, if the cellular receptors for collagen I and fibronectin probed the substrate at different frequencies, they would sense different effective values of dissipation for the same gel. A plausible hypothesis is that the association and dissociation constants of integrins with their ligands, as well as the kinetics of their coupling to the actomyosin system, could influence the frequency of cell mechanosensing. Plotnikov et al.[21] reported that the frequency of focal adhesion tugging on the substrate is 0.1 Hz. This frequency is dependent on the velocity of actin retrograde flow[22]. However, it is not known if this phenomenon takes place during the early adhesion phase, at the scale of single receptors or prior to formation of mature contractile machinery.

Attachment of adhesion molecules to both the linear and the crosslinked part of the gel led to similar average cell areas on elastic and viscoelastic gels. One possible reason for the low sensitivity to viscosity on these gels is that cells might be able to move the proteins conjugated to linear PAA to locally increase their concentration and promote adhesion, as occurs on a larger scale on viscoplastic gels[19,26,27]. PAA can be pulled further than the collagen I presented on the network (Fig. 2c), and hence cells might be able to locally re-arrange adhesion proteins at the surface of the gel. However, we were not able to observe any change in fibronectin density by confocal imaging of fluorescently labeled protein (data not shown). Another hypothesis is that the coupling of adhesion proteins to the linear PAA and the network of crosslinked PAA could result in two distinct populations of cell–matrix contacts transducing a viscoelastic or a viscous signal to the cell, and the coexistence of these signals could promote cell spreading. In the context of the model presented by Chan and Odde[22] describing substrate sensing as the result of the formation of clutches, our gels would enable clutches to be in contact with

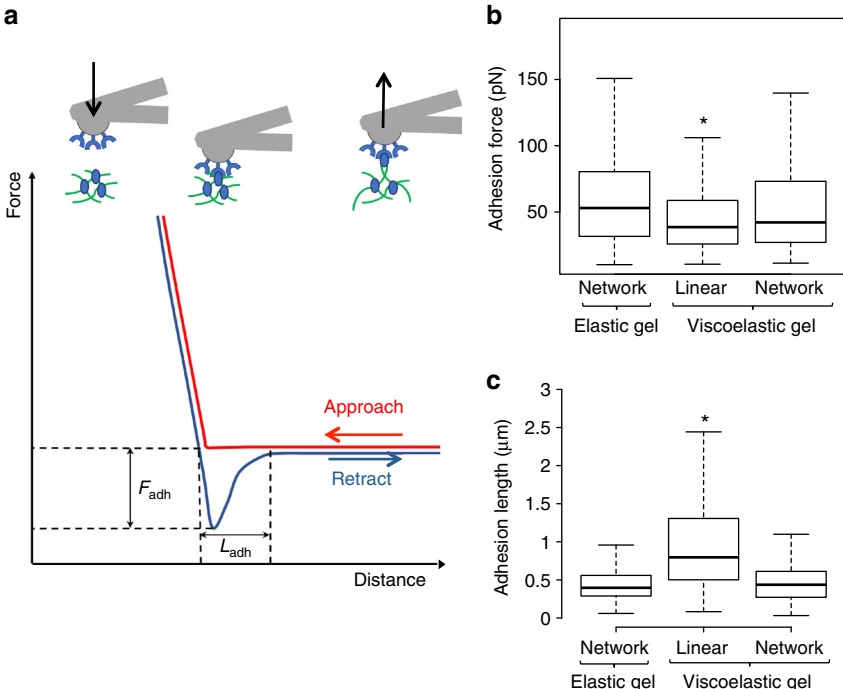

**Fig. 2** Characterization of protein crosslinking to the surface of the gel. **a** Principle of the measurement. The AFM tip is brought close to the gel until a contact is made between the anti-collagen antibody on the tip and the protein at the surface of the gel. Then the tip is retracted until the contact breaks at the values noted as $L_{adh}$ and $F_{adh}$. **b** Boxplot of the adhesion forces between the anti-collagen I antibody and the collagen I presented on the networked or the linear PAA. **c** Boxplot of the adhesion length between the anti-collagen I antibody and the collagen I presented on the networked or the linear PAA. *Statistically significant difference with a $p$-value of <0.01, as calculated by non-parametric Tukey's tests

two kinds of PAA molecules with low and high compliance through the engagement of proteins bound to either the elastic or the viscous part of the gel. This phenomenon could favor the "frictional slippage" regime of the clutches, described as a dynamic where multiple clutches are stably engaged and the mechanical heterogeneity of the cell–matrix contacts could facilitate the establishment of force bearing adhesions, thus promoting cell spreading. As the clutches would have different mechanical properties, some of them could dissipate part of the tension transmitted through the substrate and prevent the sudden increase in traction forces responsible for the disengagement of all the clutches, described as of the "load and fail" dynamic.

When adhesion molecules were attached only to the linear PAA, fibroblasts were unable to form stable attachments or spread on viscoelastic gels linked to collagen I (Fig. 3b) despite the presence of accessible collagen I molecules (Fig. 2), but were able to attach and spread on gels presenting fibronectin (Fig. 4b). For this experiment, the purely elastic gels were a negative control for cell adhesion, as these gels do not contain linear PAA molecules and only the linear PAA was functionalized. Therefore, there was no activated PAA present to crosslink proteins to the gel, and the absence of cells reflects their inability to form non-specific attachments to the inert polyacrylamide network[28]. On viscoelastic gels presenting collagen I through linear PAA, fibroblasts initially developed protrusions and formed transient adhesions with the substrate without spreading, but after 24 h no adherent cells remained on any of the gels. We conclude that fibroblasts pulling on collagen I attached to linear PAA encounter a very low elastic resistance and a high viscous dissipation preventing spreading, and the lack of tension even when integrins can bind to abundant ligands at the surface prevents stable adhesion[21,29]. On gels presenting fibronectin on the dissipative

PAA (Fig. 4, right column), 3T3 cells initially spread and then slowly retracted with many detaching from the gel, but after 24 h there were still some cells attached and spread (Fig. 4b, c), indicating that fibronectin is sufficient for adhesion. We thus conclude that permanent matrix elasticity is required for cell spreading on collagen I but not on fibronectin. This observation highlights the biochemical differences in mechanosensing by different classes of integrins engaged by different proteins crosslinked to the surface of the gel. 3T3 cells have spread areas significantly larger on weakly dissipative gels than on highly dissipative gels (Fig. 4c), suggesting that cells sense the viscoelasticity of the substrate even when attached to the viscous component.

**Viscoelastic fluids versus viscoelastic solids as matrices**. Our results differ from some other reports in the literature[11,19,26,27], likely because the materials used have significantly different mechanical properties. Cell spreading on soft viscoelastic solids with a permanent elastic modulus underlying the stress relaxation at short times differs from the response of cells to the previous dissipative substrates designed so far, which had been viscoelastic fluids with irreversible plastic deformation[11,18,19,26]. Because our gels are non-plastic, covalently crosslinked gels that cannot be remodeled by cells, integrin ligands are not able to be pulled together to make more concentrated foci, which appear to be needed to explain the larger spread area of contractile cells on viscoplastic media[11,26,30]. Local increase in collagen density could lead to enhanced cell signaling and spreading[17,19], and hence the effects of plasticity and viscous dissipations cannot be differentiated.

Altogether, our results enforce previous findings[19,26] on the importance of plasticity and matrix remodeling to promote cell

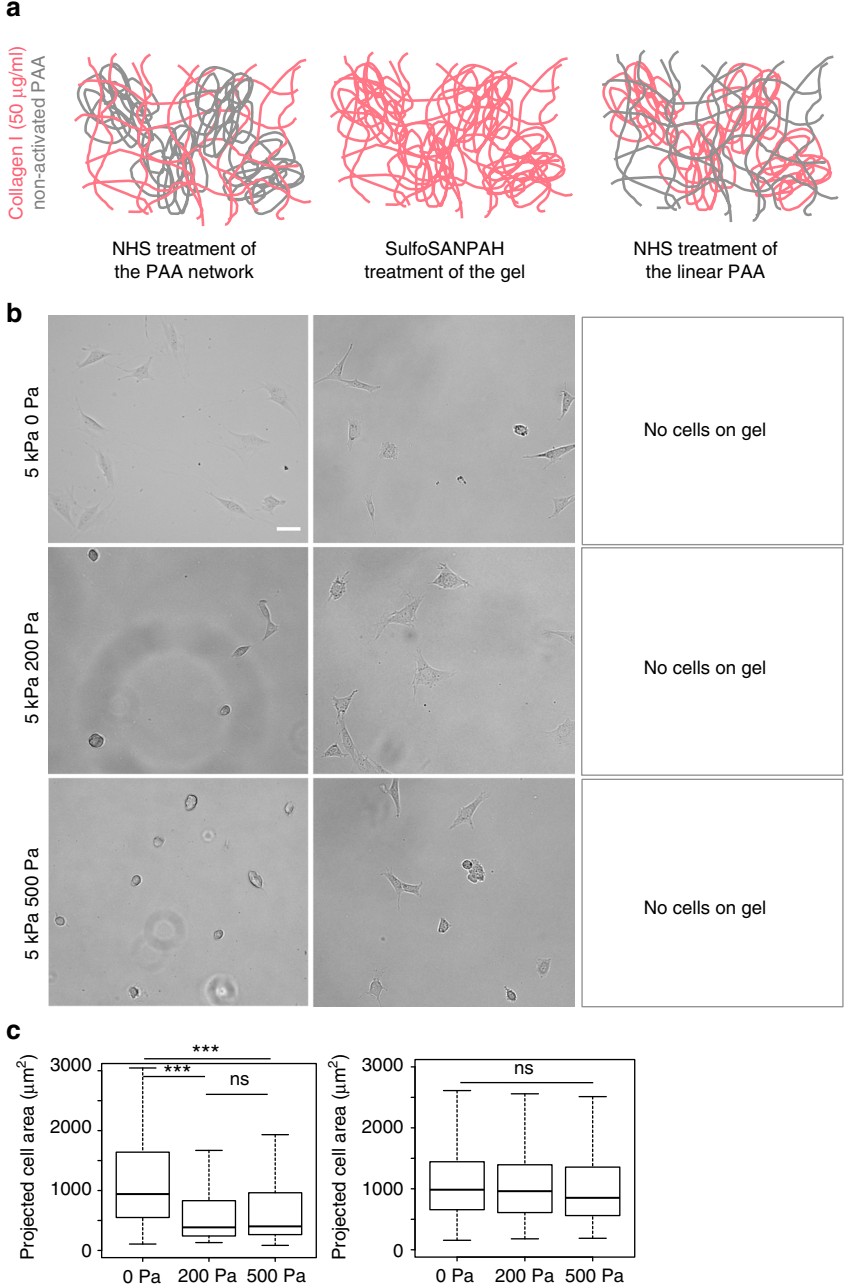

**Fig. 3** Characterization of 3T3 cell spreading on collagen I-coated gels. **a** Cartoon representing the three ways to conjugate collagen I to viscoelastic PAA gels. Adhesion proteins are crosslinked to the elastic part of the gel by adding NHS during the polymerization of the PAA network. Collagen I is crosslinked to both forms of PAA in the gel by sulfoSANPAH treatment. Collagen I is conjugated to the viscous part of the gel by adding NHS during the polymerization of the linear PAA. **b** Bright field images of 3T3 cells after 24 h of spreading on gels with G′ = 5 kPa and variable G″, coated with 50 µg/ml collagen I. Scale bar = 50 µm. **c** Average projected cell area of 3T3 fibroblasts after 24 h (right) on gels functionalized with collagen I on the elastic part of the gel (left) or on the elastic and the viscous part of the gel (right). Sample size: NHS gels 0 Pa n = 179 cells, 200 Pa n = 185 cells, 500 Pa n = 187 cells; sulfoSANPAH gels: 0 Pa n = 314 cells, 200 Pa n = 314 cells, 500 Pa n = 315 cells. ***Statistically significant difference with a p-value of <0.01, as calculated by non-parametric Tukey's tests; "ns" indicates no statistical difference

spreading and establish that viscous dissipation has a different effect from plasticity. Viscous dissipation alone tends to decrease cell spreading if it cannot be overcome by a local clustering of adhesion proteins allowed by plastic remodeling.

**Energy dissipation modifies the distribution of paxillin**. The effects of dissipation on cell spreading suggest that adhesion sites formed at the links between integrins and the substrate would be affected by change in the loss modulus G″. Paxillin is a major component of focal adhesion complexes, and its clustering is characteristic of the formation of focal adhesions. The distribution of actin and the presence of paxillin patches within fibroblasts on elastic and viscoelastic gels were visualized by immunofluorescence using confocal microscopy (Figs. 4a and 5a). Almost all of the 3T3 cells (>80%) plated on 5 kPa elastic gels exhibited an actin network organized in stress fibers (SFs) and had large paxillin patches (Figs. 5c, d and 6d, e). The

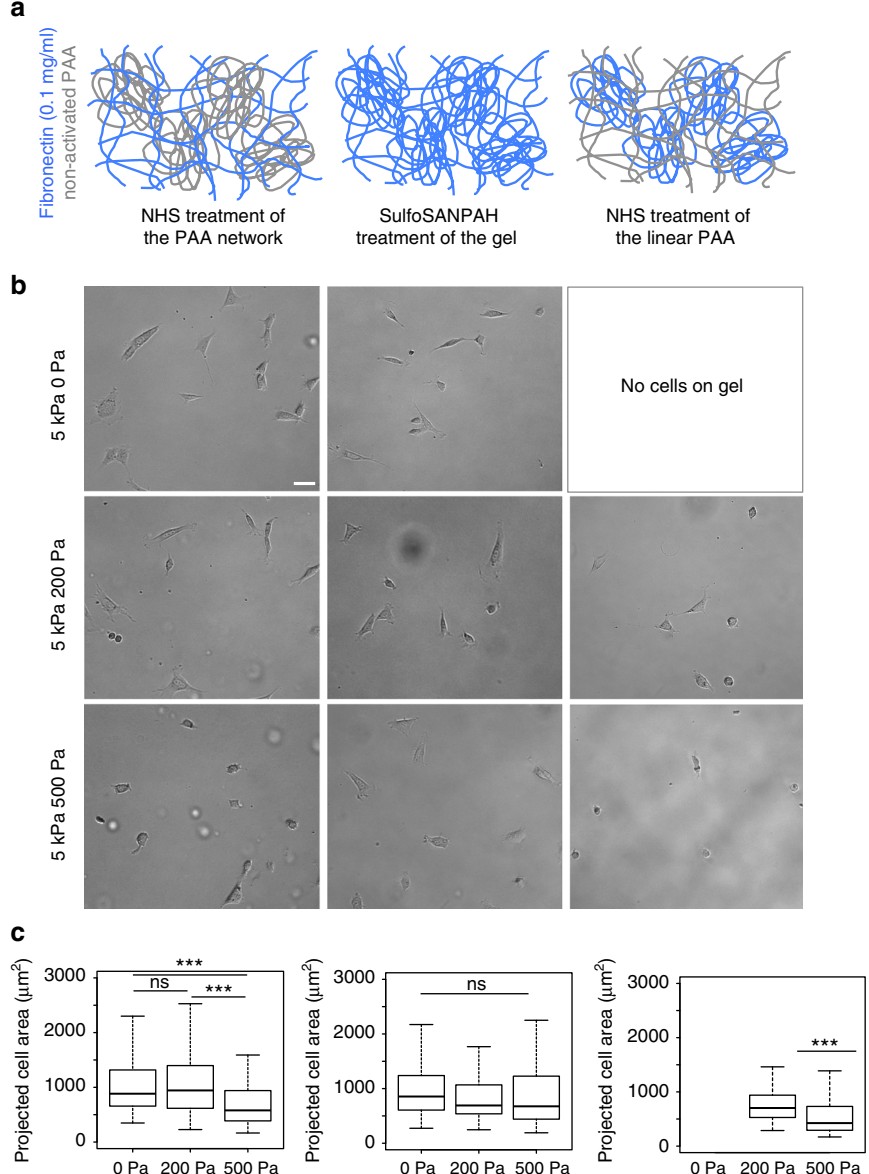

**Fig. 4** Characterization of 3T3 spreading on fibronectin-coated gels. **a** Cartoon representing the three ways fibronectin was crosslinked to viscoelastic PAA gels. **b** Bright field images of 3T3 cells after 24 h on gels with G′ = 5 kPa and variable G″ coated with 0.1 mg/ml human fibronectin. Scale bar = 50 µm. **c** Average projected cell area of 3T3 fibroblasts after 24 h on gels functionalized with fibronectin on the elastic part of the gel (left), on the elastic and the viscous part of the gel (middle) and on the linear PAA (viscous; right). Sample size: NHS gels 0 Pa $n$ = 118 cells, 200 Pa $n$ = 127 cell, 500 Pa $n$ = 104 cells; sulfoSANPAH gels: 0 Pa $n$ = 116 cells, 200 Pa $n$ = 119 cells, 500 Pa $n$ = 144 cells ***Statistically significant differences with a $p$-value of <0.01, as calculated by non-parametric Tukey's tests; "ns" indicates no statistical difference

percentages were lower on viscoelastic gels, except for the gels presenting collagen I through both the linear and the crosslinked part of the gel where more than 90% of the cells contained stress fibers and paxillin clusters. These results suggest that energy dissipation through the substrate prevents formation of large focal adhesions on viscoelastic gels. On gels presenting proteins on both the networked and the linear PAA (Figs. 5c and 6d) more cells contain stress fibers on collagen I-coated gels than on fibronectin gels. This is surprising because the cell's projected areas are comparable in these two conditions (Figs. 3c and 4c). This result highlights that establishing cell–matrix contacts through collagen I or fibronectin molecules results in different cellular responses to viscous dissipation. For all conditions, the number of cells containing SFs was similar to the percentage of

cells containing paxillin patches (Figs. 5d and 6e), suggesting that the appearance of these two cellular structures was interdependent[31].

We then quantified the number and area of paxillin patches per cell (Figs. 5e, f and 6f, g) on elastic and viscoelastic gels. Fibroblasts on gels coated with collagen I versus fibronectin exhibited differences in both the average number and area of patches per cell. On collagen I-coated gels, the number of patches per cell was approximately 15 for elastic gels and for viscoelastic gels presenting the protein on both forms of PAA. However, fibroblasts on viscoelastic gels presenting collagen I on the elastic network only formed approximately 7 patches on average (Fig. 5e). On fibronectin-coated gels, there were approximately 20 adhesion sites per cell for all conditions except when

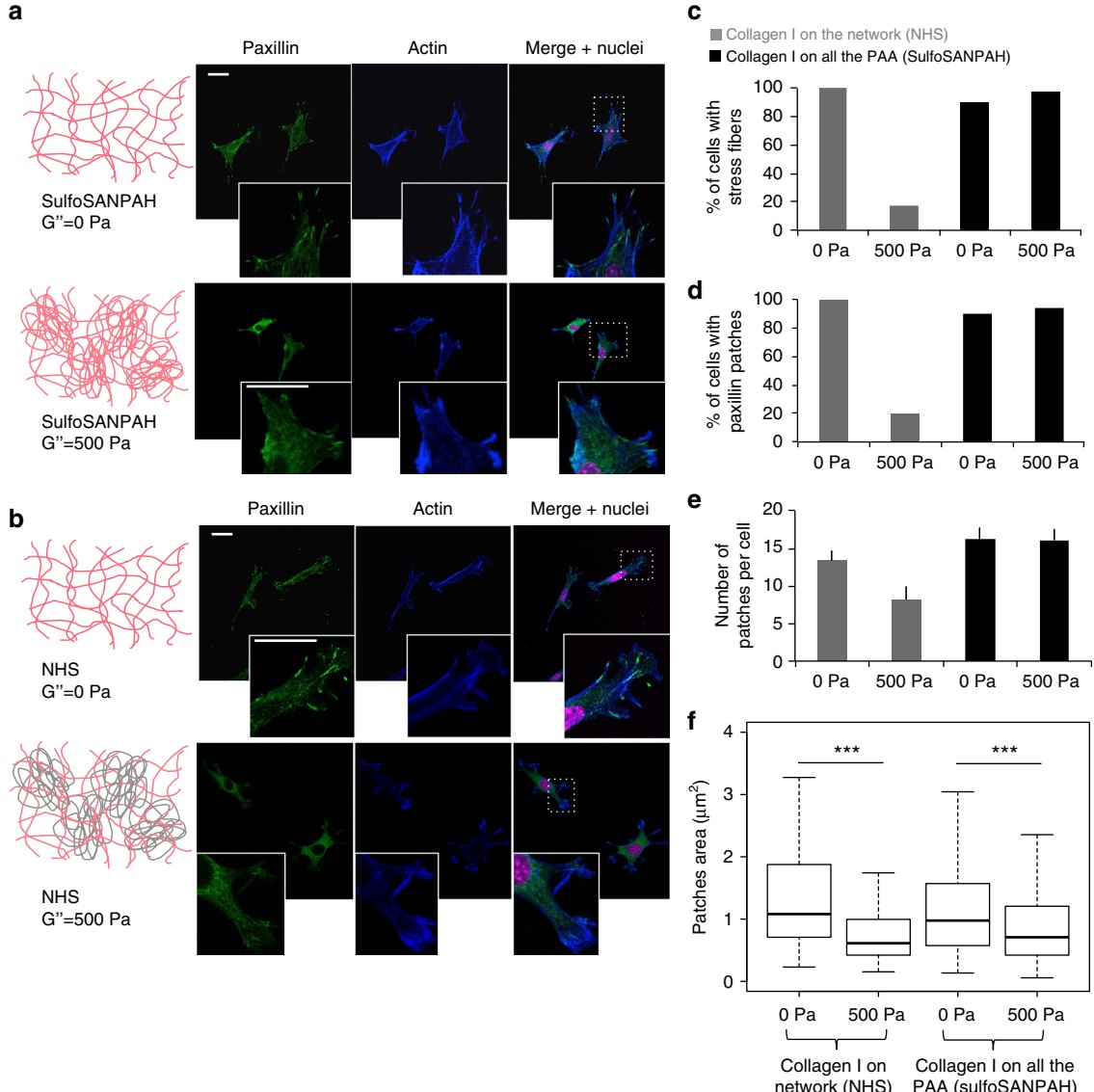

**Fig. 5** Analysis of actin stress fibers and paxillin patches of 3T3 fibroblasts incubated for 24 h on gels presenting collagen I. **a** Confocal images of paxillin (green), actin (blue) and the nucleus (pink) of 3T3 cells on 5 kPa elastic gels (top) and viscoelastic gels (bottom) with 50 μg/ml collagen I coating both forms of PAA. Scale bars = 20 μm, insets are magnification x3 of the dotted boxes. **b** Confocal images of paxillin (green), actin (blue) and nuclei (pink) of 3T3 cells incubated for 24 h on 5 kPa elastic gels (top) and viscoelastic gels (bottom), both with 50 μg/ml collagen I coating the crosslinked network of PAA. Scale bars = 20 μm, insets are magnification x3 of the dotted boxes. **c** Percentage of cells containing stress fibers on elastic and viscoelastic gels as a function of G" and the type of PAA presenting collagen I. Sample size: NHS 0 Pa $n = 8$ cells, NHS 500 Pa $n = 13$ cells, sulfoSANPAH 0 Pa $n = 24$ cells, sulfoSANPAH 500 Pa $n = 34$ cells. **d** Percentage of cells containing paxillin clusters as a function of G" and the type of PAA presenting collagen I. Sample size: NHS 0 Pa $n = 8$ cells, NHS 500 Pa $n = 13$ cells, sulfoSANPAH 0 Pa $n = 24$ cells, sulfoSANPAH 500 Pa $n = 34$ cells. **e** Average number of paxillin patches per cell as a function of the value of G" and the mode of presenting collagen I at the surface of the gel. Sample size: NHS 0 Pa $n = 8$ cells, NHS 500 Pa $n = 13$ cells, sulfoSANPAH 0 Pa $n = 24$ cells, sulfoSANPAH 500 Pa $n = 34$ cells. Error bars represent the standard error. **f** Average size of paxillin clusters as a function of the G" of the substrate and the mode of presenting collagen I to the cells. Sample size: NHS 0 Pa $n = 8$ cells, NHS 500 Pa $n = 13$ cells, sulfoSANPAH 0 Pa $n = 24$ cells, sulfoSANPAH 500 Pa $n = 34$ cells. ***Statistically significant differences with a $p$-value of <0.01, as calculated by non-parametric Tukey's tests

fibronectin was crosslinked to linear PAA, when the average was approximately 30 sites per cell. The viscosity of the substrate had no significant effect on the number of paxillin patches on fibronectin-coated gels. Viscosity resulted in a decrease in the number of clusters on viscoelastic gels with collagen I on the elastic part of the gel. Strikingly, paxillin clusters were significantly smaller on viscoelastic gels than on elastic gels in all conditions tested (Figs. 5f and 6g). This result is in agreement with the work of Cameron et al.[19] and indicates that focal

adhesion growth is impaired by energy dissipation through the substrate, independently of the adhesion protein presented at the surface of the gel.

**Viscoelasticity regulates cell stiffness and differentiation.** The stiffness of 3T3 cells plated on elastic and viscoelastic gels was measured with AFM (Fig. 7a). Fibroblasts plated on gels presenting collagen I or fibronectin on the crosslinked network were

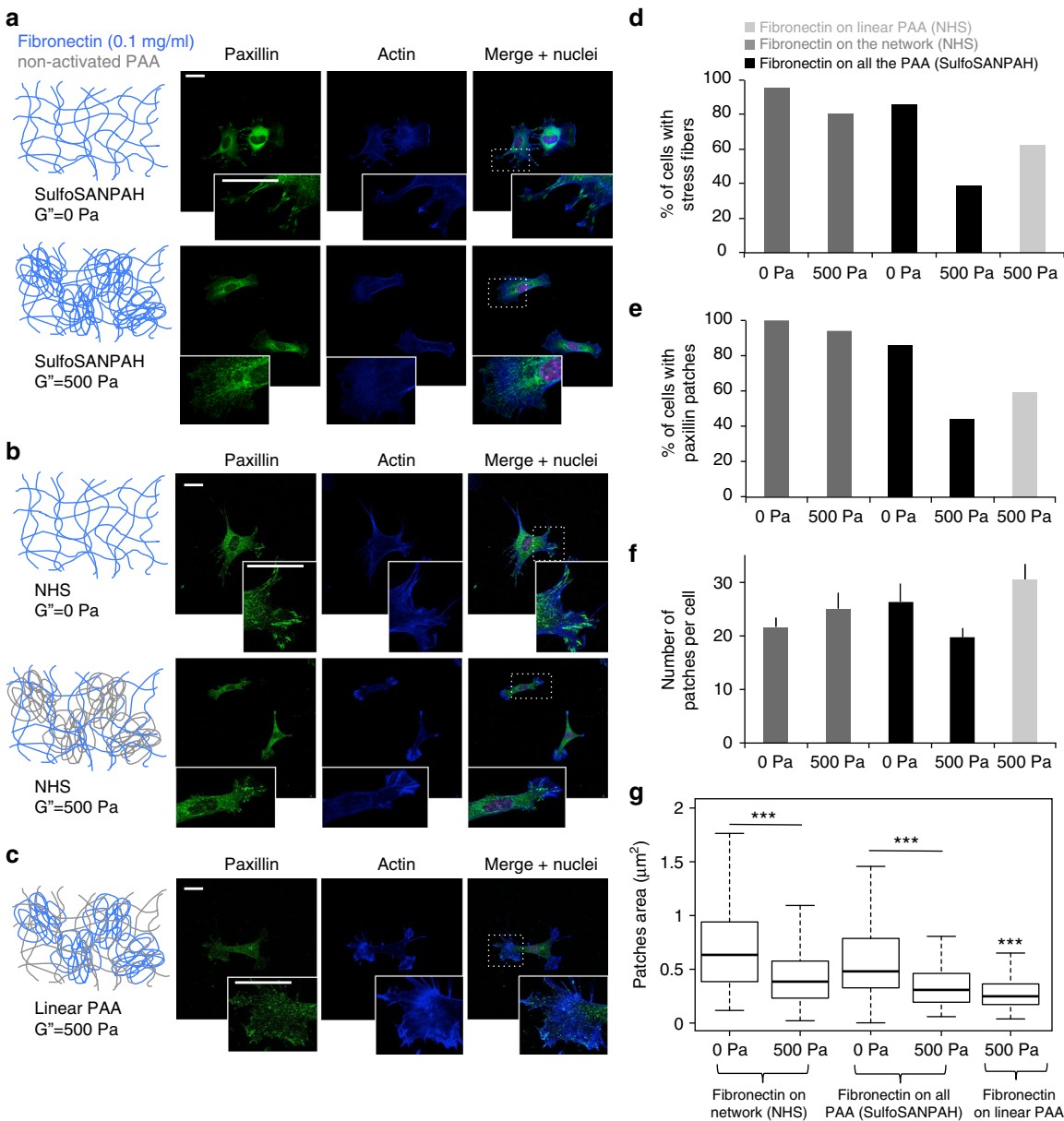

**Fig. 6** Analysis of actin stress fibers and paxillin patches of 3T3 cell incubated for 24 h on fibronectin-coated PAA gels. **a** Confocal images of paxillin (green), actin (blue) and nuclei (pink) staining of 3T3 cells sitting on an elastic gel (top) and a viscoelastic gel (bottom) coated with 100 μg/ml fibronectin on the crosslinked network and the linear PAA. Scale bars = 20 μm, insets are magnification x3 of the dotted boxes. **b** Confocal images of paxillin (green), actin (blue) and nuclei (magenta) of 3T3 cell sitting on an elastic gel (top) and a viscoelastic gel (G″ = 500 Pa, bottom) coated with 100 μg/ml fibronectin on the crosslinked network of PAA. Scale bars = 20 μm, insets are magnification x3 of the dotted boxes. **c** Confocal images of paxillin (green), actin (blue) and nuclei (magenta) of 3T3 cell on a 5 kPa viscoelastic gel presenting fibronectin on the linear PAA. Scale bars = 20 μm, insets are magnification x3 of the dotted boxes. **d** Percentage of cells containing stress fibers on elastic and viscoelastic gels as a function of G″ and the type of PAA presenting fibronectin. Sample size: NHS 0 Pa $n = 22$ cells, NHS 500 Pa $n = 20$ cells, sulfoSANPAH 0 Pa $n = 24$ cells, sulfoSANPAH 500 Pa $n = 27$ cells. **e** Percentage of cells containing paxillin clusters as a function of G″ and the type of PAA presenting fibronectin. Sample size: NHS 0 Pa $n = 22$ cells, NHS 500 Pa $n = 20$ cells, sulfoSANPAH 0 Pa $n = 24$ cells, sulfoSANPAH 500 Pa $n = 27$ cells. **f** Average number of paxillin patches by cell as a function of the value of G″ and the type of PAA presenting fibronectin. Sample size: NHS 0 Pa $n = 22$ cells, NHS 500 Pa $n = 20$ cells, sulfoSANPAH 0 Pa $n = 24$ cells, sulfoSANPAH 500 Pa $n = 27$ cells. Error bars represent the standard error. **g** Size of paxillin clusters as a function of G″ and the way to present collagen I to the cells. Sample size: NHS 0 Pa $n = 22$ cells, NHS 500 Pa $n = 20$ cells, sulfoSANPAH 0 Pa $n = 24$ cells, sulfoSANPAH 500 Pa $n = 27$ cells. ***Statistically significant differences with a $p$-value of <0.01, as calculated by non-parametric Tukey's tests

softer on viscoelastic gels. On gels presenting adhesion proteins on both linear and crosslinked PAA, the Young's modulus of cells was unaffected by the substrate viscosity. These results are consistent with our previous observation that cell spreading was affected by G″ if the cells were attached to the matrix only through the PAA network but not if the cells were anchored through both the network and the linear PAA (Figs. 4 and 5). 3T3 cells can discriminate between elastic and viscoelastic substrates while attached to the substrate through the elastic network, as both their projected area and their stiffness decrease on a dissipative substrate, but this mechanism does not involve the growth of focal adhesions.

Matrix viscosity alters the differentiation of cells. The impact of viscoelasticity on the phenotype of single cells was additionally tested using primary rat hepatic stellate cells, which lose their vitamin A-containing lipid droplets and differentiate into fibrogenic myofibroblasts in response to stiffness[32]. Cells were plated on elastic or viscoelastic PAA gels with collagen I attached to the network of crosslinked PAA. Cell differentiation was estimated through the evolution of the projected cell area. Cell spreading was followed for over a week (Fig. 7b), corresponding to the time scale for stellate cell differentiation into

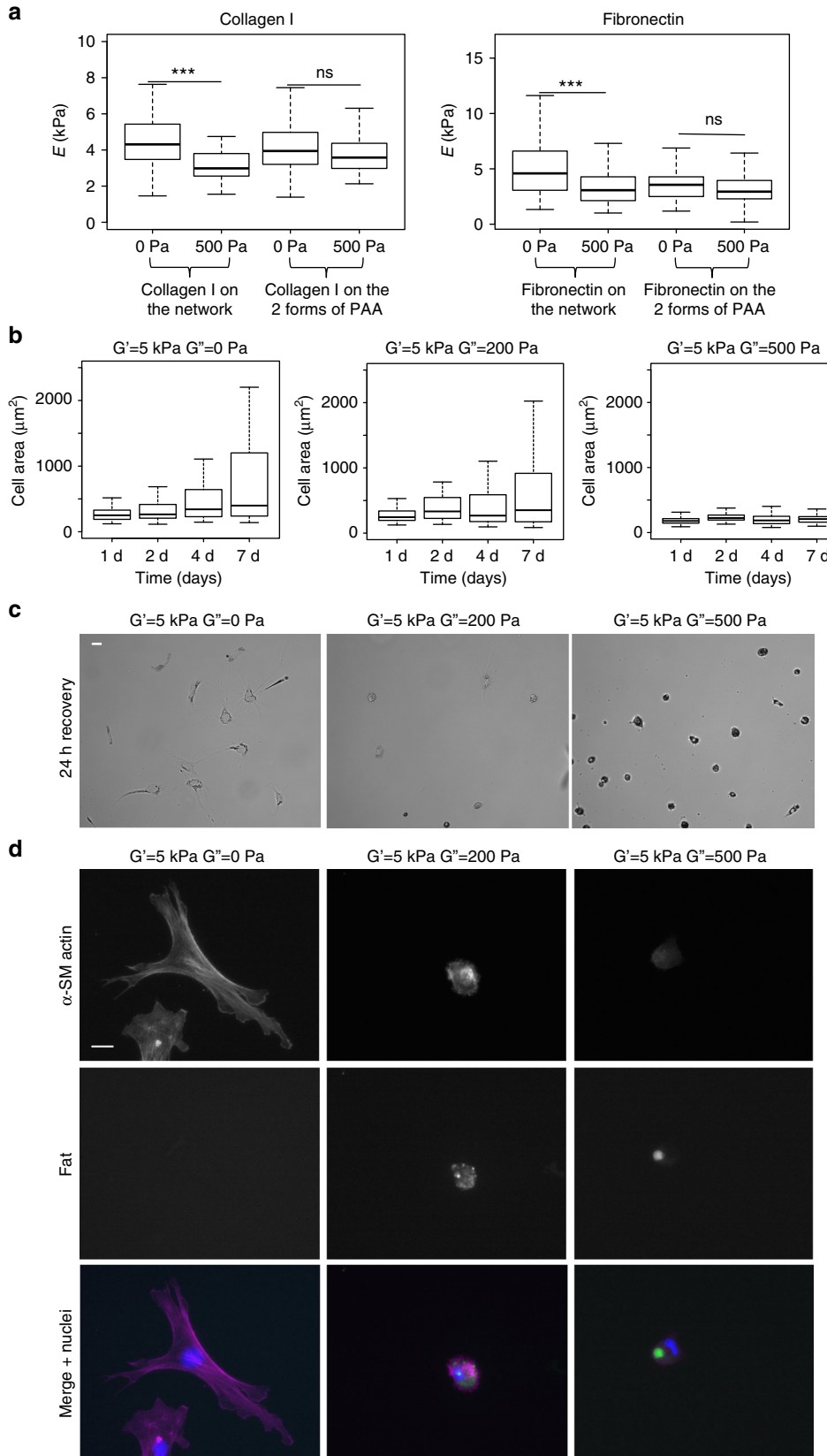

myofibroblasts on stiff substrates in vitro[32]. On elastic gels and viscoelastic gels with a low G″, the distribution of cell projected areas became broader and shifted toward higher values reflecting the differentiation of a part of the stellate cell population. The percentage of cells undergoing differentiation decreased as G″ increased, indicating that a dissipative matrix reduced the probability of differentiation even when the baseline elastic modulus, often taken as the measure of stiffness, was constant and sufficient to drive differentiation in the absence of viscosity. Myofibroblasts can dedifferentiate if their environment becomes softer[33]. For this reason, we performed a recovery experiment and assessed the impact of viscoelasticity on the morphology of established myofibroblasts. Myofibroblasts (hepatic stellate cells that had been plated on stiff plastic and allowed to differentiate for 7 days) were replated on elastic and viscoelastic gels with a stiffness of 5 kPa. Bright field images of cell morphologies after 24 h are shown in Fig. 7c, with fluorescence images in Fig. 7d. Cells on elastic gels were extensively spread and had the typical morphology of myofibroblasts, including the expression of the differentiation marker α-smooth muscle actin[32]. On moderately dissipative gels (G″ = 200 Pa), cells started to retract and re-acquired lipid droplets while still expressing α-smooth muscle actin. On highly dissipative gels (G″ = 500 Pa), cell phenotype was much closer to non-differentiated stellate cells, the staining showed that only a small amount of α-smooth muscle actin remained in the cells and the lipid droplets were restored (Fig. 7d). Taken together, these experiments with primary hepatic stellate cells demonstrate the role of energy dissipation through the substrate in cell differentiation.

## Discussion

We have developed a novel type of viscoelastic PAA hydrogel with separately controllable elastic and viscous moduli. The chemical nature and the structure of this gel allow a selective crosslinking of adhesion proteins to the elastic, the viscous or both elements of the material. These hydrogels provide a way to characterize in vitro the effect of substrate viscous dissipation on cell signaling, proliferation and differentiation. This work sheds light on the importance of time-dependent phenomena in cell mechanosensing and introduces the idea that the frequency of the molecular interactions implicated in mechanotransduction might be one of the keys to understanding the cellular "sense of touch".

The characterization of fibroblast phenotype on viscoelastic gels shows that substrate stress relaxation reduces cell stiffness and projected area when cells are attached to the crosslinked network of the gel, whereas the size of adhesion sites is always small on dissipative substrates. On gels presenting adhesion proteins on both forms of PAA, it appears that cell area is unchanged between elastic and viscoelastic gels, but the size of paxillin patches is dramatically reduced. This finding—that the

size of cell-substrate adhesion sites can be uncoupled from cell spreading—appears to be novel and not reported for cells on rigid or purely elastic substrates. Focal adhesion size is regulated by matrix stiffness and actomyosin contractility[34,35], and the small size of focal adhesions on viscoelastic substrates is in agreement with the hypothesis that low traction forces develop at the cell-substrate interface due to relaxation of the forces exerted by the cell on the substrate. The growth of focal adhesions and the associated actomyosin forces drive the differentiation of hepatic stellate cells[36] and the disruption of these two phenomenon by the substrate energy dissipation could be the origin of the reduced stellate cell differentiation we observed.

This work suggests that viscous dissipation in biological tissues is a determinant of cell phenotype and tissue homeostasis. In vivo tissue stiffness, usually quantified by a shear storage modulus or elastic Young's modulus, is known to regulate cell proliferation and differentiation[1,3,32,37], and our work now shows that the effect of elasticity can be modulated by viscosity. This effect is especially important in the context of pathologies like cancer and fibrosis[9,10,12] where changes in tissue viscoelasticity are associated with and may drive disease progression. In these disease states, not only are the elastic moduli altered, but so are the rates of stress relaxation, or the ratio of the loss modulus to the storage modulus, which are often more strongly altered than the absolute values of the elastic moduli[8–10]. Viscoelastic PAA hydrogels offer a platform to accurately reproduce the in vitro time-independent and time-dependent mechanical properties of healthy and pathological tissues.

## Methods

**Linear polyacrylamide preparation**. Linear PAA solution was generated by polymerizing 5% (w/v) acrylamide for 2 h at 37 °C. The acrylamide solution was previously degassed and polymerization was initiated by the addition of 0.024% ammonium persulfate (APS) and 0.05% tetramethylethylenediamine (TEMED). The resulting viscous solution was stored at 4 °C in the dark.

In order to crosslink adhesion proteins only to the viscous part of the gel, the linear polyacrylamide molecules need to be activated. A total volume of 0.4% AA-NHS in toluene was added to the 5% acrylamide solution which was then agitated thoroughly, incubated at room temperature (RT) for 5 min and centrifuged for 5 min at 1000×g. Toluene was removed and polymerization was initiated as described above. The amount of activated linear PAA incorporated to the G″ = 200 Pa and G″ = 500 Pa gels was identical to insure similar adhesion protein densities on the gel. In order to keep a value of G″ = 500 Pa, non-activated linear PAA was also added to the highly viscous gels.

**Polyacrylamide gel preparation and protein coating**. Viscoelastic gels were obtained by polymerizing a branched network of PAA in a solution of linear PAA. The linear PAA was already fully polymerized; in consequence, it could not interact with acrylamide monomers and be crosslinked to the network. The viscosity of the gel depends on the amount of linear PAA incorporated into the network. Elastic PAA gels were made following the same protocol but without linear poly-acrylamide. Acrylamide, bis-acrylamide and linear PAA were mixed and degassed. Polymerization was initiated by adding APS and TEMED. Gels were polymerized 30 min at 25 °C. To covalently attach collagen I to the elastic network of PAA, 0.4%

---

**Fig. 7** Analysis of the effect of viscoelasticity on 3T3 cell stiffness and hepatic stellate cell differentiation. **a** Young's modulus of 3T3 fibroblasts as a function of G″. Cells were incubated for 24 h on 5 kPa elastic (G″ = 0 Pa) and viscoelastic (G″ = 500 Pa) gels coated with collagen I (left) or fibronectin (right). Sample size: Collagen I gels NHS 0 Pa n = 76 cells, NHS 500 Pa n = 68 cells, sulfoSANPAH 0 Pa n = 112 cells, sulfoSANPAH 500 Pa n = 104 cells; fibronectin gels NHS 0 Pa n = 187 cells, NHS 500 Pa n = 176 cells, sulfoSANPAH 0 Pa n = 149 cells, sulfoSANPAH 500 Pa n = 139 cells. **b** The evolution over 7 days of the projected area of hepatic stellate cells on elastic (left) and viscoelastic gels (middle and right). The distribution of cell area values becomes broader over time. This shift toward greater values is characteristic of the differentiation of part of the cell population into myofibroblasts. Sample size: 5 kPa 0 Pa gel day 1 n = 114 cells, day 2 n = 117 cells, day 4 n = 117 cells, day 7 n = 70 cells; 5 kPa 200 Pa gel day 1 n = 112 cells, day 2 n = 108 cells, day 4 n = 119 cells, day 7 n = 114 cells; 5 kPa 500 Pa gel day 1 n = 113 cells, day 2 n = 95 cells, day 4 n = 107 cells, day 7 n = 106 cells. **c** Bright field images of myofibroblasts incubated for 24 h on elastic and viscoelastic PAA gels coated with collagen I. Myofibroblasts have a smaller area on viscoelastic gels; on the gels with G″ = 500 Pa, cell morphology is similar to that of undifferentiated cells. Scale bar = 50 μm. **d** Fluorescence images of myofibroblasts incubated for 24 h on elastic and viscoelastic PAA gels coated with collagen I. Nuclei (blue) are stained with Hoechst, α-smooth muscle actin (magenta) has been stained by immunofluorescence and lipid droplets (green) were stained with Bodipy® 493/503. Scale bar = 20 μm. ***Statistically significant differences with a p-value of <0.01, as calculated by non-parametric Tukey's tests; "ns" indicates no statistical difference

AA-NHS in toluene was added to the gel mix prior to polymerization and incubated for 5 min at RT. The toluene was removed by centrifugation for 5 min at 1200 RCF and polymerization initiated as previously described. Then, gels were rinsed 3 times in 50 mM HEPES pH = 8.2 to remove the residual acrylamide monomers.

In order to crosslink adhesion molecules to both the networked and linear PAA, the two types of PAA were polymerized as described above and were coated with a sulfoSANPAH 0.1 mM solution in 25% dimethyl sulfoxide and 75% HEPES 50 mM, pH = 8.2. Activation of the PAA was triggered by exposing the sulfoSANPAH-immersed gels to ultraviolet light for 8 min. The sulfoSANPAH was removed by rinsing 3 times with HEPES 50 mM, pH = 8.2.

Rat tail collagen I (Corning) was crosslinked to the activated PAA by incubating the gels in a 0.05 mg/ml solution of collagen I in HEPES 50 mM, pH = 8.2 for 2 h at RT. The pH was carefully controlled in order to prevent the formation of collagen I fibers at the surface of the gel. Gel functionalization with fibronectin extracted from human plasma was performed in a 0.1 mg/ml solution of fibronectin in HEPES 50 mM, pH = 8.2, for 2 h at RT. All the gels used in this study presented only one type of adhesion protein at their surface. The gels were rinsed 3 times with PBS and incubated with Dulbecco's modified Eagle's medium (DMEM) for 30 min at 37 °C before cell plating. 3T3 cells were put in suspension by 5 min of incubation with trypsin (GIBCO) at 37 °C, then centrifuged to remove the trypsin and seeded on gels in DMEM.

**Gel rheology.** Gel rheology was carried out with Kinexus and Bohlin stress controlled rheometers (Malvern), and a RFS3 strain controlled rheometer (TA Instruments), using 40 mm and 8 mm circular parallel plate geometries, respectively. The PAA solution was deposited on the bottom plate of the rheometer immediately after initiation of polymerization. Then, the upper plate was lowered to a 1 mm gap to form a disk between the two plates. Polymerization was followed by applying an oscillating torque to the gel at 1 rad/s and 2% strain typically for 30 min until G' and G" values reached a plateau. Once the polymerization was complete, the stress relaxation of the gel was tested for 10 min. The frequency dependence of the gel was characterized with 2% strain from 0.05 Hz to 50 Hz.

**AFM measurements of collagen–anti-collagen interactions.** AFM force spectroscopy was performed with a NanoWizzard4 AFM microscope (JPK Instruments, Germany). The silicon nitride cantilevers (Novascan Technologies, USA) had a nominal spring constant equal to 0.07–0.072 N/m and a 1 μm diameter bead attached to the tip extremity. Cantilevers were cleaned in chloroform and then functionalized by incubating dried tips with APTMS (3-aminopropyl)trimethoxysilane) under a desiccator for 1 h. Subsequently, tips were immersed in 2.5% glutaraldehyde in dH₂O for 30 min and washed 3 times in dH₂O to remove the excess glutaraldehyde. Finally, tips were incubated with 0.1 mg/ml anti-collagen I antibody in PBS (EMD Millipore, USA) for 30 min at 37 °C and were washed 3 times in PBS. Functionalized cantilevers were kept at 4 °C in PBS and used within 2 days.

Hydrogel samples coated with human collagen I (Advanced BioMatrix) were probed with an extension rate of 1 Hz, a retraction rate of 0.25 Hz and a 0.2 s waiting time at the surface with relative set point equal to 0.3 nN. Force versus distance curves were collected within a square of 5 μm × 5 μm at the 10 different locations. The spring constant of each cantilever was calibrated by collecting thermal resonance curves (cantilever fluctuations as a function of frequency) prior to the experiment. Data were analyzed using JPK data processing software which allows the determination of the unbinding force (force of adhesion ($F_{adh}$)) and the unbinding length (length of adhesion event ($L_{adh}$)) for the collagen I–anti-collagen interaction. Force curves were analyzed individually and manually. Each set of data included a control blocking experiment. Then, 0.1 mg/ml anti-collagen I antibody in PBS was added to the gel at the end of the measurement and force curves were collected for 1 h without rinsing the antibody. Blocking of collagen I presented on top of the gels allows determining the contribution for non-specific interactions between the tip and the gel surface. In response to blocking the number of single specific unbinding events was significantly reduced (Supplementary Figure 4).

**Cell culture.** 3T3 mouse fibroblasts from the ATCC were maintained in DMEM (Gibco) with 10% fetal bovine serum (FBS, ATCC) and 1% streptomycin and penicillin (S/P, Gibco) at 37 °C and 5% CO₂.

Primary hepatic stellate cells were extracted from retired breeder Sprague-Dawley adult male rats. All experiments involving animals were approved by the University of Pennsylvania Institutional Animal Care and Use Committee and followed the guidelines set out by the US Public Health Service Policy on the Humane Care and Use of Laboratory Animals.

The liver was perfused in situ with pronase I and then with collagenase II[32]. Cells were isolated by density gradient centrifugation in 9% Nycodenz (Sigma) solution at 1400×g. Primary cells were maintained in Medium 199 (Gibco) supplemented with 10% FBS (ATCC), L-glutamine (Gibco), Amphotericin B (Gibco) and 1% S/P (Gibco). For recovery experiments, stellate cells were cultivated on plastic for 7 days to induce differentiation into myofibroblasts and were then trypsinized and replated on PAA gels in medium containing 25 μM of all-trans

retinol (Sigma). Cell areas were manually traced for at least 50 cells over at least three independent experiments.

**Immunofluorescence, staining and microscopy.** 3T3 cells and myofibroblasts were fixed 24 h after being plated on gels. Cells were incubated with paraformaldehyde 4% for 30 min, permeabilized with Triton 0.5% in PBS for 15 min and then saturated with 1% serum albumin bovine for 30 min at RT. Immunostaining was performed with primary anti-paxillin monoclonal rabbit antibodies in a 1:250 dilution (Abcam ab32084) or primary anti-α-smooth actin antibody in a 1:200 dilution (Abcam ab7817). Secondary antibodies were Alexa Fluor goat anti-rabbit 488 in a 1:1000 dilution (Life Technologies) or Alexa Fluor goat anti-mouse 568 in a 1:1000 dilution (Life Technologies). Lipid droplets were stained with Bodipy® 493/503 (Molecular Probes). Cells were then incubated with Phalloidin Alexa Fluor 647 (Molecular Probes) and Hoechst (Molecular probes) for 1 h according to the manufacturer's instructions.

Cells were imaged with an inverted Zeiss LSM 710 confocal microscope through a 40 × water objective or with a Leica II inverted microscope with a 40× air objective. Paxillin patches were manually counted and circled on the confocal images.

**Atomic force microscopy: cell stiffness measurement.** AFM elasticity measurements were conducted using a DAFM-2× Bioscope (Veeco, Woodbury, NY) mounted on an Axiovert 100 microscope (Zeiss, Thornwood, NY). Cell indentation was made with silicon nitride cantilevers characterized by spring constant equal 0.07–0.08 N/m (Novascan Technologies, Ames, IA) and SiO₂ bead of 1 μm in diameter attached. All cantilevers used in the study were commercially pre-calibrated. Before each measurement, samples were washed two times in growing medium and the measurements were conducted at RT for no longer than 1 h per sample. Only single cells were measured and each cell was indented three times at the endoplasmic region outside of the nucleus with a constant rate equal to 1 Hz.

Elasticity measurements were realized by recording the so-called "force versus indentation" curves that were later analyzed in the frame of Hertz model, assuming spherical geometry of the cantilever tip. In order to calculate the Young's modulus, the first 30% of every "force versus indentation" curve was analyzed and fitted to previously described formula[3]. The Poisson ratio was assumed to be equal to 0.5 and final indentation depth did not exceed 500 nm.

**Statistics.** The distribution of the projected cell area and cell stiffness values was tested for normality with the Shapiro–Wilk test. The distribution of our data sets did not follow a Gaussian law, and hence we used non-parametric statistics tests to compare cells on elastic and viscoelastic substrates. The Kruskal–Wallis test was applied to compare three data sets, and the Mann–Whitney test was used for the comparison of two data sets. Differences were considered to be statistically significant when $p$-values were <0.01. Statistically similar data sets were indicated by "ns".

**Data availability.** All relevant data are available from the authors.

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

## Acknowledgements

The authors would like to thank the National Institutes of Health (R01EB017753 and CA193417) and the NSF Center for Engineering Mechanobiology (CMMI-154857) for funding this research. Thanks to Anne Van Oosten, Katrina Cruz, Jessica Llewellyn and LiKang Chin for their help and wise insights and to Shang-You Tee for early discussions about viscoelastic substrates.

## Author contributions

E.E.C. and K.P. designed, performed and analyzed the experiments. P.A.J. performed the dynamic light scattering and analyzed the viscometry data. E.E.C., K.P., R.G.W. and P.A.J. designed the project and wrote the manuscript.

## Additional information

**Competing interests:** The authors declare no competing financial interests.

