## [Peer Review File · Nature Communications]

Reviewers' comments:

Reviewer #1 (Remarks to the Author):

This manuscript by Charier et al. presents an approach to prepare viscoelastic materials for studying the effect of substrate energy dissipation on cell morphology and differentiation. While some of the results are interesting, the current version of the manuscript might not be suitable for publication in Nature Communications.

There are two major issues with this manuscript.

First, the findings on cellular responses to viscoelastic matrix cues presented in this manuscript contradict the findings from other published papers on the same research topic (for example, ref. 11 and 14 cited by the authors). In those papers, the effect of substrate energy dissipation (by using very distinct material platforms) is reported to enhance cell spreading and differentiation. However, the findings in this manuscript show the entirely opposite trend. While the authors have characterized the differences on focal adhesions on both their elastic and viscoelastic gels, their discussions remain on a phenomenological level. In addition, the authors state that the size of focal adhesion can be decoupled from cell morphology (based on results from Figures 4 and 5), but it is unclear whether such differences originate from the effect of substrate viscoelasticity, or the differences in the way how proteins are chemically tethered. Without providing further evidences to support their claims or providing deeper discussions on the possible reasons that lead to contradicting results, it is difficult to rationalize their findings.

Second, the materials characterizations are not adequately performed to support the authors' claims.

- a. Because the linear polyacrylamide (PAA) polymers are important for preparing the viscoelastic gels, their synthesis and characterizations should be carefully reported. The information of molecular weight and distribution of linear PAA is lacking in the methods section.
- b. A lot of assumptions or parameters in estimating the diffusive motions of PAA in gels are not justified, including the assumption of a theta solvent condition (line 63), hydrodynamic radius (line 61), Kuhn length (line 62), gel mesh size (line 66). Many of these parameters should be calculated based on theories, or measured experimentally.
- c. The authors should experimentally validate that the linear PAA polymers do not escape the gels over the time period of the cellular experiments. This can be done, for example, by fluorescently labeling the linear PAA and measuring the changes of fluorescent intensity of gels and solvents over time.
- d. It is unclear why the authors only report the moduli at a frequency of 0.16 Hz, even though the frequency sweep (Figure 1F) suggests that G'' is highly dependent on frequency. It is also unclear how the changes in PAA concentration alters the frequency profile of G'' for different gel formulations.
- e. The authors use two different methods to conjugate adhesive proteins onto the elastic or viscoelastic substrates. When adhesive proteins are conjugated in both forms of PAAs, it is possible that the mobility of linear PAA is drastically reduced because the adhesive protein can potentially crosslink both forms of PAAs. If this is true, it is expected that the mechanical properties of gels change after modification. In addition, the authors hypothesize that there are two forms of PAA and adhesive proteins (Page 6, lines 151-163), can it be tested experimentally?
- f. The heterogeneity of the gels should be characterized, especially for the ones with linear PAA. Based on the information in Table 1, and the schematics in the figures throughout the manuscript, it seems that the authors imply some level of structural heterogeneity in their gels. If this is true, do the authors expect a heterogeneous distribution of adhesive proteins that might confound their findings?

Other issues:

- (1) Page 4, line 89, the statement that "the frequency of the cellular mechanisms probing the

substrate determines the value of G'' felt by cells" is confusing. Does it imply that the cells do not feel the elastic component? Also, previous studies (for example, ref. 11 and 14) suggest that cells integrate the mechanical cues over time, instead of simply reading mechanical cues at a specific frequency.

(2) The stress relaxation experiment shown in Figure 1C is performed in the nonlinear regime. Also, the relaxation time constant seems to be less than 1 s, not 100 s stated by the authors.

(3) The description of using activated acrylic acid monomers to prepare linear PAAs is not accurate. It is a copolymerization process (Page 4). The acrylic acid NHS esters do not activate the amide bonds of linear PAA.

(4) Page 6, line 149, the authors should provide the data showing no changes in fibronectin density.

(5) When comparing the effect of different proteins, it is unclear whether the differences come from the chemical differences or the tethering density. In addition, the authors hypothesize that cellular receptors for collagen I and fibronectin probe the substrate at different frequencies, could the authors provide related references?

(6) The information in Table 1 does not match the information in Figure 1. For example, the formulation of 4% PAA in Figure 1 gives $G'' = 500$ Pa, whereas the linear PAA concentration reported in Table 1 is 2.75%.

(7) Page 6, line 176, figure 2 should read figure 3.

(8) Page 7, line 178, figure 2 should read figure 3.

(9) Page 9, line 254, G'' should be 500 Pa, not 600 Pa.

Reviewer #2 (Remarks to the Author):

This paper presents some very interesting results suggesting cellular response to viscosity. The most valuable contribution of this paper is the development of a novel gel system in which viscosity can be tuned independent of storage modulus in a manner that allows for ligand presentation on one or both. In an effort to characterize cellular response to these materials in a rigorous manner, unfortunately, quite a few puzzling results are presented – these results are not sufficiently explained to warrant publication of this manuscript in Nature Comm.

1. In terms of gel preparation, description of NHS vs sulfoSANPAH activation is not sufficiently clear. The authors state, "the copolymerization of AA-NHS with acrylamide during preparation of the linear PAA enabled us to obtain activation of the linear PAA after polymerization. The "co" and "after" seem conflicting. Moreover, I do not see why the NHS approach could not have been used to activate both network and linear PAA eliminating the need for a second type of activation.

2. Further comparison to work of Chaudhuri and co-workers, Cameron and co-workers, and others who have probed similar effects should be included. Even if these results were complicated by matrix remodeling or similar issues, some discussion of the relevant findings and possible reasons for differences is seen needed.

3. The introduction needs citations regarding the relevance of materials with this sort of G'/G'' ratio to real tissue. I am aware that some reconstituted ECM proteins certainly have ratios in the 10-20 regime discussed by the authors, but is this true in vivo where much more covalent cross-linking is generally present? References are definitely needed and some description of which tissues have these types of ratios would be helpful.

4. The hypotheses and conclusions related to cells sensing at particular frequencies is intriguing. The authors state "the frequency of the cellular mechanisms probing the substrate determines the value of G'' felt by cells" – so what frequencies are these? Even if the answers are unknown, the authors can speculate – what are the relevant interactions and the time scales of those? Perhaps

integrin activation and/or recycling time scales would be relevant? This comment is relevant not only for the sentence stated above but also for page 5 where the authors hypothesize why collagen and fibronectin results may be different – integrin activation times may be different for the relevant collagen/fibronectin integrins.

5. I find the results in Fig. 2 and 3 fairly confounding. Initially, I expected that the cells would not feel and respond to the viscosity of the substrate if the cells can not adhere to it through ligands. This appears not to be the case, as on collagen cells only feel the viscosity when they the linear PAA is unfunctionalized. This result is similar on the fibronectin gels. This immediately brings up a question that as far as I can tell the authors do not address: is it possible that differences in presenting ligand density drive these differences in cell response? In other words, in the just-network ligand attached gels, maybe the cell area decreases with increasing linear PAA b/c the ligand density decreases as viscosity increases. Meanwhile this does not happen on the doubly treated gels since there ligand density is constant. It seems important to clarify that that is not what is driving the differences if indeed it is not. Another possibility that I do not see addressed anywhere relates to the possibility of the cells stabilizing the functionalized linear PAA. In this case the viscoelastic gels would be effectively elastic when they have ligands – if the cells attach to the ligands on the linear PAA, they rigidify the linear PAA and stabilize the system. One can rationalize many of the results through this line of thinking. The way the authors do attempt to rationalize the results, through invoking the Chan/Odde model, should be explained more clearly; what would be the expected result of frictional slippage vs. load and fail and why?

6. The authors do not appear to address the fact that gels with collagen with elastic vs. visco and elastic functionalization and fibronectin with elastic vs. visco and elastic functionalization have opposite behaviors in terms of stress fibers and paxillin patches. In other words, having the functionalized linear PAA induces an increase in stress fibers and paxillin patches with collagen coating but a decrease with fibronectin. In both cases these changes seen between the two 500Pa G'' gels are greater than changes between viscoelastic and plain elastic networks. As such it should be discussed. The result is especially surprising given that the functionalization of linear PAA on the 500Pa G'' gels leads to somewhat enhanced cell spreading for both collagen and fibronectin substrates.

7. The authors would strengthen their results on cell differentiation if they used known markers or looked at gene expression of these cells rather than just commenting on morphology.

8. In Figure 1, the authors show a frequency dependence, but do not specify which gel is being discussed. Indeed, it would be interesting to see if the frequency dependence of the G'' of the two viscoelastic gels is the same or different.

9. Figure 4F is mistakenly labeled as having fibronectin instead of collagen.

10. Figure 5D would be more readily comparable to Fig 4C if the ordering were changed and the first column was put last.

11. The legends for Fig 4C and 5D are confusing. How can collagen or fibronectin be on "2 forms of PAA" when some of these (0 Pa) have no linear PAA? I think the authors mean NHS vs. SANPAH functionalized gels for the 0 Pa gels not "on the network" vs "on 2 forms of PAA."

Response to Reviewers' comments:

Reviewer #1 (Remarks to the Author):

This manuscript by Charier et al. presents an approach to prepare viscoelastic materials for studying the effect of substrate energy dissipation on cell morphology and differentiation. While some of the results are interesting, the current version of the manuscript might not be suitable for publication in Nature Communications.

There are two major issues with this manuscript.

First, the findings on cellular responses to viscoelastic matrix cues presented in this manuscript contradict the findings from other published papers on the same research topic (for example, ref. 11 and 14 cited by the authors). In those papers, the effect of substrate energy dissipation (by using very distinct material platforms) is reported to enhance cell spreading and differentiation. However, the findings in this manuscript show the entirely opposite trend. While the authors have characterized the differences on focal adhesions on both their elastic and viscoelastic gels, their discussions remain on a phenomenological level. In addition, the authors state that the size of focal adhesion can be decoupled from cell morphology (based on results from Figures 4 and 5), but it is unclear whether such differences originate from the effect of substrate viscoelasticity, or the differences in the way how proteins are chemically tethered. Without providing further evidences to support their claims or providing deeper discussions on the possible reasons that lead to contradicting results, it is difficult to rationalize their findings.

We agree that our results are different from responses reported by other groups, but that is because previous dissipative substrates have been designed to be viscoelastic fluids, and to permit plastic deformation. Our dissipative substrates are viscoelastic solids, with an equilibrium elastic modulus that prevents irrecoverable plastic deformation. Therefore, although the responses of cells to the substrates we make are different from those reported in other studies, our data do not contradict them. Rather they point to the difference between cellular responses to visco-plastic materials (or viscoelastic fluids) and to viscoelastic solids. Each type of dissipative materials will be a more appropriate mimic of different soft tissues depending on how those native materials relax at long times. In order to explain this distinction, we added a section to the paper comparing our results with results of other groups who developed other dissipative materials for cell culture. We emphasize that the difference between our results and the one of Chaudhuri et al. and Cameron et al., in particular, originates from the nature of the material and its resulting inherent mechanical properties. We developed a viscoelastic solid that cannot be remodeled by cells while other groups worked with viscoelastic liquids that exhibit some plastic behavior. We would like to thank the reviewer for this suggestion, since comparing our results with the findings of other groups could lead us to identify more precisely the distinct contributions of plasticity and viscous dissipation to cell spreading.

We would also like to address the following concern: "In addition, the authors state that the size of focal adhesion can be decoupled from cell morphology (based on results from Figures 4 and 5), but it is unclear whether such differences originate from the effect of substrate viscoelasticity, or the differences in the way how proteins are chemically tethered."

The observed decreased size of focal adhesion on viscoelastic substrate is similarly observed for all types of viscoelastic substrates, no matter whether adhesion proteins are presented on the networked PAA, the linear PAA or both forms of PAA and using two different chemistries of attachment. For

this reason it seems unlikely that this finding can be attributed to changes in the presentation of proteins at the surface of the gel. The conclusion that focal adhesion size can be decoupled from cell morphology refers to the fact that on viscoelastic substrates the size of focal adhesions is always smaller than on elastic substrates and is thus not correlated to the projected area of the cell. We now also show data from atomic force microscopy that confirm that beads coated with anti-collagen antibodies engage their ligands similarly on both elastic and viscoelastic substrates. There is no evidence that engagement of collagen by integrins is any more sensitive to conformation than recognition of collagen by a specific antibody to it.

As described by Trappmann *et al.*¹ tethering is a phenomenon specific to collagen I, which can form fibers at the surface of a substrate. Cell adhesion and spreading can be influenced by the length of these collagen fibers. When we designed our material, we minimized the influence of tethering with 2 strategies:

- First, we carefully controlled the pH of the protein coating buffer to prevent collagen polymerization and ensure the crosslinking of monomers of collagen I at the surface of the gel. Thus the mechanical feedback perceived by the cells is not due to the formation of collagen fibers on top of the gel, but to the mechanical properties of the substrate.

- Second, we confirmed the effect of viscous dissipation on cell spreading by repeating the experiment with fibronectin (Fig. 3), which cannot form fibers at the surface of the gel. We can then be sure that cells are attached to monomers of fibronectin directly connected to PAA.

We also performed AFM pulling measurements to assess the nano-scaled mechanical properties of the gel and confirmed that the presentation of collagen I on the networked PAA is similar between elastic and viscoelastic gels. The results of these experiments have been added to the main text.

Second, the materials characterizations are not adequately performed to support the authors' claims.

a. Because the linear polyacrylamide (PAA) polymers are important for preparing the viscoelastic gels, their synthesis and characterizations should be carefully reported. The information of molecular weight and distribution of linear PAA is lacking in the methods section.

Per the reviewer's suggestion we performed additional characterization of the linear PAA and included the results in the revised manuscript. We used viscometry to determine intrinsic viscosity of the polymer and thereby calculate the average molecular weight of the linear PAA (Supporting Information Figure 1A). We characterized the size distribution of the linear PAA molecules with dynamic light scattering (S.I. Fig. 1B). The results of these two independent methods to determine polymer size give consistent results.

b. A lot of assumptions or parameters in estimating the diffusive motions of PAA in gels are not justified, including the assumption of a theta solvent condition (line 63), hydrodynamic radius (line 61), Kuhn length (line 62), gel mesh size (line 66). Many of these parameters should be calculated based on theories, or measured experimentally.

The following assumption has been removed from the article "assumption of a theta solvent condition (line 63), Kuhn length (line 62)". We replaced it with the results of viscometry and dynamic light scattering measurements. We also take into account that the culture medium, which is a good solvent for polyacrylamide, leads to an effect that would further increase the radius of gyration of the linear PAA and retard its diffusion out of the network. This conclusion is based on the finding that the relation between intrinsic viscosity of linear polyacrylamide and molecular wt, as calculated by the Mark-Houwink relation differs by less than 10% for the polymer in pure water and water with 1M NaCl².

The hydrodynamic radius of the linear acrylamide molecule was measured by dynamic light scattering. Details concerning the use of dynamic light scattering to characterize the linear PAA have been added to the supporting information “Methods” section. We apologize for omitting this information in the original manuscript.

c. The authors should experimentally validate that the linear PAA polymers do not escape the gels over the time period of the cellular experiments. This can be done, for example, by fluorescently labeling the linear PAA and measuring the changes of fluorescent intensity of gels and solvents over time.

We have now fluorescently labeled linear acrylamide and quantified the exchange of fluorescence between the gel and the solvent. However, the results are not simply related to the question of how fast the linear polymer in our substrates can leave the gel. First, the presence of fluo-acrylate interferes with the linear acrylamide polymerization by quenching the free radical polymerization reaction leading to shorter chains than when prepared only with unlabeled acrylamide, as determined by DLS. This translates as a decrease in the viscosity of the linear acrylamide solution and by a decrease in the size of the linear polymer.

Second, the fluorescence of fluo-acrylate is different in a polymerized gel and in solution; the fluorescence is about 30% lower in a polymerized gel than in an unpolymerized solution of the same formulation. The direct consequence of this is that the amount of fluorescence in the solvent is over-estimated when the fluorescence ratio between the gel and the supernatant is calculated.

We still characterized the evolution of the fluorescence repartition for 2 concentrations of fluorescein-acrylate in the linear PAA.

After 24h, we quantified the amount of fluorescence initially contained in the gel that diffused out. For gels made with linear acrylamide containing 1.10⁻⁷ g/L of fluo-acrylate, 15% of the total

fluorescence is in the solvent. For gels containing a concentration of $2 \cdot 10^{-7}$ g/L of fluo-acrylate in the linear PAA, 20 % of the fluorescence had diffused out of the network.

After 7 days, the amount of fluorescence in the solvent is still about 15% for the lowest concentration of fluo-acrylate, and is approximately 25% for the highest concentration. Thus, most of the diffusion occurred in the first 24h of gel immersion. As the presence of fluorescein interferes with the polymerization, it could be monomers or short oligomers of fluo-acrylate that diffuse out of the network. Also because the size distribution of the linear chains is large, the smallest linear PAA molecules diffuse out, while the longest chains stayed trapped in the network, and the long chains contribute most to the viscous dissipation within the gel.

In order to assess whether the diffusion affects the viscoelastic properties of the gel, gels were polymerized on a glutaraldehyde-treated coverslip and soaked in PBS for a week. G'' and G' were measured after 24h and 7 days in solution. These data are presented in Supplementary Information Figure 2. Gels exhibit a 10% decrease in the value of their G'' after one week. This is not statistically significant and within the error of this kind of measurement ($\sim 10\%$); however, we expect that the value of G'' slightly decreases over time due to the diffusion of the linear PAA out of the network since the linear chains are not chemically bound and will eventually diffuse out, but not significantly during the time course of our cell studies.

d. It is unclear why the authors only report the moduli at a frequency of 0.16 Hz, even though the frequency sweep (Figure 1F) suggests that G'' is highly dependent on frequency. It is also unclear how the changes in PAA concentration alters the frequency profile of G'' for different gel formulations.

Two studies point out that the relevant frequency for single cell mechanosensing is in the range of 0.1Hz (one event every ten seconds)^{3,4}, which corresponds to the frequency at which cells tug on their substrate and gauge the mechanical resistance of the substrate. In consequence, we choose to report the moduli of the gels at 1rad/s (0.159Hz), as it's a biologically relevant frequency. This point has been added in the main text, but we recognize that the time course of mechanosensing is unknown, and the 0.1Hz hypothesis is one of several. The frequency sweeps for all the gel formulations have been added in the supporting information (S.I. Figure 2).

e. The authors use two different methods to conjugate adhesive proteins onto the elastic or viscoelastic substrates. When adhesive proteins are conjugated in both forms of PAAs, it is possible that the mobility of linear PAA is drastically reduced because the adhesive protein can potentially crosslink both forms of PAAs. If this is true, it is expected that the mechanical properties of gels change after modification. In addition, the authors hypothesize that there are two forms of PAA and adhesive proteins (Page 6, lines 151-163), can it be tested experimentally?

To assess if the attachment of adhesion proteins to the gels has an influence on G'' , we polymerized gels and performed the coating procedure on the gels. We measured G' and G'' of gels crosslinked to collagen I with the two coating methods. As presented in S.I. Figure 3, the mechanical properties of the gels presenting collagen I are similar to the uncoated gels. Additionally, we carefully controlled the pH of the solution in which the collagen I is cross-linked to the PAA in order to prevent the formation of collagen I fibers and thus prevent the formation of bound between the linear PAA molecules.

There are two forms of PAA, linear and networked PAA, in the viscoelastic gels. Adhesion protein can be attached to the network of PAA or the linear PAA, and we hypothesized that adhesions molecules would be similarly accessible on both forms but would present different mechanical resistance, depending on the form of PAA to which they are attached. To address the reviewer's

comment, we experimentally tested and showed, with AFM pulling experiments, that the adhesion molecules attached to one or another PAA present distinct characteristics (figure 2). The results of this experiment have been added to the main text.

f. The heterogeneity of the gels should be characterized, especially for the ones with linear PAA. Based on the information in Table 1, and the schematics in the figures throughout the manuscript, it seems that the authors imply some level of structural heterogeneity in their gels. If this is true, do the authors expect a heterogeneous distribution of adhesive proteins that might confound their findings? In order to test the heterogeneity of the gels, we performed a nano-scaled stiffness mapping with AFM at an indentation depth of $1\mu\text{m}$ (S.I. Fig. 1C and 1D). At low indentation, the presence of linear PAA slightly increases the gel stiffness heterogeneity, in comparison to gels made only by a network. However, the repartition of the local stiffness measurements for elastic and viscoelastic gels overlap quite well for 85% of the forces curves. Only the 15% of extreme values are more dispersed for the viscoelastic gels. This broader distribution of local stiffness is not affecting the overall stiffness of the gel; the Young's moduli of the elastic and the viscoelastic gels are strictly identical. As a result, this heterogeneity is not likely to affect cell mechanosensing as the integration of the local stiffness signals at the scale of a whole cell will lead to similar overall stiffness sensed on elastic and viscoelastic gels.

Other issues:

(1) Page 4, line 89, the statement that “the frequency of the cellular mechanisms probing the substrate determines the value of G' ” felt by cells” is confusing. Does it imply that the cells do not feel the elastic component? Also, previous studies (for example, ref. 11 and 14) suggest that cells integrate the mechanical cues over time, instead of simply reading mechanical cues at a specific frequency.

In general it is not clear whether cells probes resistance to traction stress statically or dynamically. In principle however they do it, they seem to integrate the response to their traction forces over some period of time, which can be probed either by looking at stress relaxation or frequency dependence. We have reworded to text to be careful not to imply a specific mechanism or to conclude that cells are sensitive only to G' or G'' ; it is unlikely to be that simple. Adding a viscous component to the elastic gel produces a frequency dependence in both G' and G'' . The frequency dependence of G'' is stronger than that of G' , but both depend on the presence of the linear chains. What can be concluded from this study is that cells feel the resistance of their substrate in a frequency or time-dependent manner and the time scale of this dependence is governed by the dissipative structures in the substrate. It seems likely that on a molecular level what counts is how force or displacement changes after the cell applies a traction force to the substrate.

We have tried to clarify this issue in the text.

(2) The stress relaxation experiment shown in Figure 1C is performed in the nonlinear regime. Also, the relaxation time constant seems to be less than 1 s, not 100 s stated by the authors.

A new experiment has been done for the data in figure 1C, which now shows the stress relaxation performed in the linear regime, with a 10% strain. The results are the same.

The relaxation time constant is about 10s. This has been corrected.

(3) The description of using activated acrylic acid monomers to prepare linear PAAs is not accurate. It is a copolymerization process (Page 4). The acrylic acid NHS esters do not activate the amide bonds of linear PAA.

The text has been rephrased to state precisely that not all the monomers are activated, and that the activated monomers copolymerize with the non-activated monomers.

The reviewer is correct that the acrylic acid does not activate the amide bonds; we had meant that the active NHS replaces nonreactive amide in the polymer subunit. This text has been removed..

(4) Page 6, line 149, the authors should provide the data showing no changes in fibronectin density.

We were not able to quantify the fibronectin density on linear PAA for the two formulations of viscoelastic gels. We tried to image fluorescently labeled fibronectin, and to immunostain the fibronectin at the surface of the gel, but we were not able to quantify the amount of fluorescent protein bound to the surface of the gel, as previously reported by other groups trying this method^{1,5}

However, while making the gels we made sure to incorporate the exact same amount of activated linear PAA in the two formulations, so in theory, the number of crosslink site for fibronectin will be similar on low and high viscosity gels. This is detailed in the “Method” section.

(5) When comparing the effect of different proteins, it is unclear whether the differences come from the chemical differences or the tethering density. In addition, the authors hypothesize that cellular receptors for collagen I and fibronectin probe the substrate at different frequencies, could the authors provide related references?

We do not believe that tethering differences are to take in account as we carefully design the gels to not be sensitive to tethering changes, as shown by the AFM pulling experiment presented in figure 2. Additionally, the procedure to cross-link proteins at the surface of the gels is identical, so the protein density and the residues through which attachment occurs should be similar for collagen I and fibronectin. Also, since these are only PAA gels, the types of complications in the way proteins are tethered for example to PDMS, plastic, or glass are not relevant, since except for the covalent bond there is no significant binding of the protein to the substrate. However, fibronectin and collagen might induce a different cellular response and adhesion strength at similar concentrations⁶⁻⁸. As a consequence, we think that the differences come from the chemical nature of the ligands and their receptors.

We are only speculating that cellular receptors for collagen I and fibronectin might probe the substrate at different frequencies. We emphasize in the text (page 7, lines 179-185) that this is not a proven hypothesis, so there is no reference available.

(6) The information in Table 1 does not match the information in Figure 1. For example, the formulation of 4% PAA in Figure 1 gives $G'' = 500$ Pa, whereas the linear PAA concentration reported in Table 1 is 2.75%.

The table presented the absolute amount of linear PAA, while the Fig. 1 showed the % of the solvent replaced by linear PAA. This has been corrected and the absolute amount of linear PAA appears now in both table 1 and figure 1.

(7) Page 6, line 176, figure 2 should read figure 3. Yes, this has been corrected.

(8) Page 7, line 178, figure 2 should read figure 3. Yes, this has been corrected

(9) Page 9, line 254, G'' should be 500 Pa, not 600 Pa. Yes, this is fixed.

Reviewer #2 (Remarks to the Author):

This paper presents some very interesting results suggesting cellular response to viscosity. The most valuable contribution of this paper is the development of a novel gel system in which viscosity can be tuned independent of storage modulus in a manner that allows for ligand presentation on one or both. In an effort to characterize cellular response to these materials in a rigorous manner, unfortunately, quite a few puzzling results are presented – these results are not sufficiently explained to warrant publication of this manuscript in Nature Comm.

1. In terms of gel preparation, description of NHS vs. sulfoSANPAH activation is not sufficiently clear. The authors state, “the copolymerization of AA-NHS with acrylamide during preparation of the linear PAA enabled us to obtain activation of the linear PAA after polymerization. The “co” and “after” seem conflicting. Moreover, I do not see why the NHS approach could not have been used to activate both network and linear PAA eliminating the need for a second type of activation.

The description of the polyacrylamide preparation and activation has been corrected to be more precise.

The AA-NHS activation could have been used in all cases. We had to make a choice between using the same kind of activation in all the gel conditions and using a newly polymerized solution of linear PAA at each experiment, or using the same linear acrylamide stock solution in the gels presenting protein on the network and performing activation with AA-NHS and SulfoSANPAH. We choose to use two kinds of activation because it allows us to use the same stock solution of linear polyacrylamide for the gels presenting protein on the network only, and on both forms of PAA. Thus, the mechanical properties of the gels, and especially their viscosities, are more reproducible between the experiments presenting proteins on the network. To work around the use of two crosslink methods, we choose to make two elastic gels as controls, one for each crosslink technique. So we can compare cell behavior on elastic and viscoelastic gels made of PAA activated by the same method. In addition SulfoSANPAH is much more widely used than AA-NHS for soft substrates, and therefore easier to relate to other studies.

2. Further comparison to work of Chaudhuri and co-workers, Cameron and co-workers, and others who have probed similar effects should be included. Even if these results were complicated by matrix remodeling or similar issues, some discussion of the relevant findings and possible reasons for differences is seen needed.

A section comparing our work to the work of the Chaudhuri and the Cameron groups has been added to the main text. Thank you for this suggestion; by further comparing these experimental systems we highlight the distinct contributions of viscous dissipation and plastic remodeling to cell behavior on viscoelastic material.

3. The introduction needs citations regarding the relevance of materials with this sort of G''/G' ratio to real tissue. I am aware that some reconstituted ECM proteins certainly have ratios in the 10-20 regime discussed by the authors, but is this true in vivo where much more covalent cross-linking is generally present? References are definitely needed and some description of which tissues have these types of ratios would be helpful.

We added citations to prove the relevance of the G''/G' ratio for in vivo tissues. These kinds of ratios are reported in the literature for in vivo tissues such as brain, spinal cord, fat and liver.

4. The hypotheses and conclusions related to cells sensing at particular frequencies is intriguing. The

authors state “the frequency of the cellular mechanisms probing the substrate determines the value of G' felt by cells” – so what frequencies are these? Even if the answers are unknown, the authors can speculate – what are the relevant interactions and the time scales of those? Perhaps integrin activation and/or recycling time scales would be relevant? This comment is relevant not only for the sentence stated above but also for page 5 where the authors hypothesize why collagen and fibronectin results may be different – integrin activation times may be different for the relevant collagen/fibronectin integrins.

Two studies point out that the relevant frequency is in the range of 0.1Hz. This is emphasized in the main text. We also provide more information on the mechanisms that could define frequencies relevant to cell sensing. (See response to point d from reviewer #1.)

5. I find the results in Fig. 2 and 3 fairly confounding. Initially, I expected that the cells would not feel and respond to the viscosity of the substrate if the cells can not adhere to it through ligands. This appears not to be the case, as on collagen cells only feel the viscosity when they the linear PAA is unfunctionalized. This result is similar on the fibronectin gels. This immediately brings up a question that as far as I can tell the authors do not address: is it possible that differences in presenting ligand density drive these differences in cell response? In other words, in the just-network ligand attached gels, maybe the cell area decreases with increasing linear PAA b/c the ligand density decreases as viscosity increases. Meanwhile this does not happen on the doubly treated gels since there ligand density is constant. It seems important to clarify that that is not what is driving the differences if indeed it is not. Another possibility that I do not see addressed anywhere relates to the possibility of the cells stabilizing the functionalized linear PAA. In this case the viscoelastic gels would be effectively elastic when they have ligands – if the cells attach to the ligands on the linear PAA, they rigidify the linear PAA and stabilize the system. One can rationalize many of the results through this line of thinking. The way the authors do attempt to rationalize the results, through invoking the Chan/Odde model, should be explained more clearly; what would be the expected result of frictional slippage vs. load and fail and why?

Concerning the following statements:

“in the just-network ligand attached gels, maybe the cell area decreases with increasing linear PAA b/c the ligand density decreases as viscosity increases”

The amount of activated polymer is determined by the amount of AA-NHS and is the same in the elastic and the viscoelastic gels presenting protein only on the network, as both have a network made of 8% acrylamide. So the resulting amount of protein attached to the gels should be similar. To confirm this, we performed AFM pulling experiments (presented in Fig. 2) proving that the length and force of adhesion are similar for collagen molecules presented on the network of both elastic and viscoelastic gels. In other words, the ligand density is similar on these gels.

“Another possibility that I do not see addressed anywhere relates to the possibility of the cells stabilizing the functionalized linear PAA”.

We do not think that the cells could make the gel effectively elastic because this would require them to crosslink the linear acrylamide in 3 dimensions; the cells can perform crosslinks only at the cell-matrix interface. Additionally, if the cells were making the linear PAA effectively elastic, the area of cells plated on gels presenting fibronectin on the linear acrylamide only should increase with the amount of linear PAA in the gel. We can see on Fig. 4 that the cell area decreases when the amount of linear PAA, and thus the viscosity, increases.

The fact that cells pull on the linear acrylamide probably influences cell spreading by locally changing the distribution of adhesion proteins. The measure of the length of adhesion by AFM pulling (Fig. 2) shows that the degree of freedom of the molecules attached to the linear acrylamide is significantly increased. In addition, when cells are plated on viscoplastic gels where collagen is attached only to the linear chains they initially adhere, showing that they engage the collagen, but within minutes to hours release from the substrate, presumably because in this time they either pull the chain fully or totally out of the network, thereby losing resistance to traction force.

“The way the authors do attempt to rationalize the results, through invoking the Chan/Odde model, should be explained more clearly; what would be the expected result of frictional slippage vs. load and fail and why?”

Details concerning the explanation of our results in the context of this model have been added to the manuscript and we have rephrased this part of the manuscript for clarity.

6. The authors do not appear to address the fact that gels with collagen with elastic vs. visco and elastic functionalization and fibronectin with elastic vs. visco and elastic functionalization have opposite behaviors in terms of stress fibers and paxillin patches. In other words, having the functionalized linear PAA induces an increase in stress fibers and paxillin patches with collagen coating but a decrease with fibronectin. In both cases these changes seen between the two 500Pa G⁰ gels are greater than changes between viscoelastic and plain elastic networks. As such it should be discussed. The result is especially surprising given that the functionalization of linear PAA on the 500Pa G⁰ gels leads to somewhat enhanced cell spreading for both collagen and fibronectin substrates.

It is true that these gels have opposite behavior in terms of the % of cells containing stress fibers and paxillin patches. The average number of patches/cell, however, is similar for cells on gels presenting collagen I or fibronectin on both the elastic and the viscous PAA, at about 20 patches / cell. The difference in stress fibers is mentioned in the text page 10 line 295 “The percentages were lower on viscoelastic gels, except for the gels presenting collagen I through both the linear and the crosslinked part of the gel where more than 90% of the cells contained stress fibers and paxillin clusters”. We discussed further this change in the percentage of cell containing stress fibers in line 260-264 on page 9.

We agree that the decrease in stress fibers on fibronectin coated gels is surprising because it does not correlate with the cell spreading data. However, cell spreading is not enhanced on viscoelastic gels presenting protein on both forms of PAA, it is just similar to cell spreading on an elastic substrate of the same stiffness. Discussion about this point as been added in the main text.

7. The authors would strengthen their results on cell differentiation if they used known markers or looked at gene expression of these cells rather than just commenting on morphology.

We repeated the studies of hepatic stellate cell differentiation and added fluorescence images of hepatic stellate cells stained for two markers: α -smooth actin, which is specifically expressed in myofibroblastic (differentiated) stellate cells, and lipid droplets, which are specifically stored in non-myofibroblastic (quiescent) hepatic stellate cells.

8. In Figure 1, the authors show a frequency dependence, but do not specify which gel is being discussed. Indeed, it would be interesting to see if the frequency dependence of the G⁰ of the two viscoelastic gels is the same or different.

We thank the reviewer for this comment. The figure caption now states that the frequency sweep of Figure 1 is performed on a viscoelastic gel with $G''=500\text{Pa}$. The frequency dependence plots of all the gels have been added to the supplementary information section.

9. Figure 4F is mistakenly labeled as having fibronectin instead of collagen.
Yes, this mistake has been corrected.

10. Figure 5D would be more readily comparable to Fig 4C if the ordering were changed and the first column was put last.

The order of the column has been changed to make the plots of Fig. 5 and Fig.6 easily comparable.

11. The legends for Fig 4C and 5D are confusing. How can collagen or fibronectin be on “2 forms of PAA” when some of these (0 Pa) have no linear PAA? I think the authors mean NHS vs. SANPAH functionalized gels for the 0 Pa gels not “on the network” vs. “on 2 forms of PAA.”

Yes, the reviewer makes a good point and the legend has been changed to avoid confusion.

Bibliography

1. Trappmann, B. *et al.* Extracellular-matrix tethering regulates stem-cell fate. *Nat. Mater.* **11**, 642–649 (2012).
2. Munk, P., Aminabhavi, T. M., Williams, P., Hoffman, D. E. & Chmelir, M. Some Solution Properties of Polyacrylamide. *Macromolecules* **13**, 871–876 (1980).
3. Plotnikov, S. V., Pasapera, A. M., Sabass, B. & Waterman, C. M. Force fluctuations within focal adhesions mediate ECM-rigidity sensing to guide directed cell migration. *Cell* **151**, 1513–1527 (2012).
4. Chan, C. E. & Odde, D. J. Traction dynamics of filopodia on compliant substrates. *Science* **322**, 1687–1691 (2008).
5. Wen, J. H. *et al.* Interplay of matrix stiffness and protein tethering in stem cell differentiation. *Nat. Mater.* **13**, 979–987 (2014).
6. Lacouture, M. E., Schaffer, J. L. & Klickstein, L. B. A comparison of type I collagen, fibronectin, and vitronectin in supporting adhesion of mechanically strained osteoblasts. *J. Bone Miner. Res. Off. J. Am. Soc. Bone Miner. Res.* **17**, 481–492 (2002).
7. Colombo, E., Calcaterra, F., Cappelletti, M., Mavilio, D. & Bella, S. D. Comparison of Fibronectin and Collagen in Supporting the Isolation and Expansion of Endothelial Progenitor Cells from Human Adult Peripheral Blood. *PLOS ONE* **8**, e66734 (2013).
8. Bouafsoun, A., Othmane, A., Kerkeni, A., Jaffrézic, N. & Ponsonnet, L. Evaluation of endothelial cell adherence onto collagen and fibronectin: A comparison between jet impingement and flow chamber techniques. *Mater. Sci. Eng. C* **26**, 260–266 (2006).

REVIEWERS' COMMENTS:

Reviewer #2 (Remarks to the Author):

The authors performed additional experiments and clarified aspects of the manuscript. As such, it is now acceptable for publication.

REVIEWERS' COMMENTS:

Reviewer #2 (Remarks to the Author):

The authors performed additional experiments and clarified aspects of the manuscript. As such, it is now acceptable for publication.

The referee did not raise any issues on the current version of the paper. In consequence we did not modified the content of the article. However we performed modifications of the text to comply with the editorial requests.